# Autophagy and the Insulin-like Growth Factor (IGF) System in Colonic Cells: Implications for Colorectal Neoplasia

**DOI:** 10.3390/ijms24043665

**Published:** 2023-02-11

**Authors:** Aldona Kasprzak

**Affiliations:** Department of Histology and Embryology, University of Medical Sciences, Swiecicki Street 6, 60-781 Poznan, Poland; akasprza@ump.edu.pl; Tel.: +48-61-854-6441; Fax: +48-61-854-6440

**Keywords:** colonic and rectal epithelial cells, colorectal cancer, autophagy, insulin-like growth factor (IGF) system, tumor microenvironment (TME) cells

## Abstract

Colorectal cancer (CRC) is one of the most common human malignancies worldwide. Along with apoptosis and inflammation, autophagy is one of three important mechanisms in CRC. The presence of autophagy/mitophagy in most normal mature intestinal epithelial cells has been confirmed, where it has mainly protective functions against reactive oxygen species (ROS)-induced DNA and protein damage. Autophagy regulates cell proliferation, metabolism, differentiation, secretion of mucins and/or anti-microbial peptides. Abnormal autophagy in intestinal epithelial cells leads to dysbiosis, a decline in local immunity and a decrease in cell secretory function. The insulin-like growth factor (IGF) signaling pathway plays an important role in colorectal carcinogenesis. This is evidenced by the biological activities of IGFs (IGF-1 and IGF-2), IGF-1 receptor type 1 (IGF-1R) and IGF-binding proteins (IGF BPs), which have been reported to regulate cell survival, proliferation, differentiation and apoptosis. Defects in autophagy are found in patients with metabolic syndrome (MetS), inflammatory bowel diseases (IBD) and CRC. In neoplastic cells, the IGF system modulates the autophagy process bidirectionally. In the current era of improving CRC therapies, it seems important to investigate the exact mechanisms not only of apoptosis, but also of autophagy in different populations of tumor microenvironment (TME) cells. The role of the IGF system in autophagy in normal as well as transformed colorectal cells still seems poorly understood. Hence, the aim of the review was to summarize the latest knowledge on the role of the IGF system in the molecular mechanisms of autophagy in the normal colon mucosa and in CRC, taking into account the cellular heterogeneity of the colonic and rectal epithelium.

## 1. Introduction

Colorectal cancer (CRC) is one of the most common human malignancies worldwide. In 2020, it was in third place in terms of detection and accounted for 10% of all cancers detected in both sexes. As a cause of death from all cancers, CRC ranked second in both sexes [1]. Most cases of CRC are sporadic and are the result of factors related to a poor lifestyle that leads to obesity or metabolic syndrome (MetS). Genetic/epigenetic changes in CRC cause dysregulation of multiple signaling pathways leading to tumor development, growth and progression [2].

One of the pathways that plays an important role in the pathogenesis and progression of CRC is the insulin-like growth factor (IGF) system [3,4]. The role of the IGF system components including insulin, IGF-1 and IGF-2 (IGFs), IGF-1 type I receptor (IGF-1R), insulin receptor (IR), IGF binding proteins (IGF BPs) and insulin receptor substrate 1/2 (IRS-1/IRS-2) in cell proliferation/apoptosis and the development of resistance to both chemotherapeutic drugs and agents based on mutation of the epidermal growth factor receptor (EGFR) are well known and widely reported [3,4,5,6]. The hyperactivation of IGF-1R causes primary and secondary resistance to EGFR inhibition in *KRAS* wild-type (WT) metastatic CRC (mCRC) through the upregulation of phosphatidylinositol-3-kinase (PI3K)/protein kinase B (PKB, or AKT) signaling. Interestingly, underlying the resistance of mCRC to treatment with anti-EGFR monoclonal antibodies (mAbs) (e.g., Cetuximab and Panitumumab) is the dysregulation of cellular autophagy [7,8,9].

Autophagy or type II programmed cell death is defined as a multistep, catabolic process, comprising the degradation of proteins and/or DNA aggregates, as well as aberrant cellular organelles and/or microbes, resulting in the uptake by autophagosomes and digestion in lysosomes to sustain cellular metabolism. Eventually, autophagy protects normal cells from neoplastic transformation by removing damaged proteins, DNA, cellular structures and reducing reactive oxygen species (ROS) [10,11,12,13,14]. The process of autophagy in carcinogenesis depends on many factors (e.g., tissue type, stage of disease, driver mutations and metabolic sensitivity), and plays an even paradoxical role [13,14,15,16]. At the stage of tumor transformation, autophagy inhibits tumor growth (anti-tumoral autophagy role), while in established disease under conditions of metabolic stress, it has an oncogenic function (pro-tumoral autophagy role) [10,13,16,17,18]. Transformed cells may thus activate autophagy in response to cellular stress and/or increased metabolic demands associated with increased cell proliferation [13,18].

The role of autophagy in the maintenance of normal intestinal homeostasis has been raised in the literature in recent years, and the effects of dysregulation of this process in relation to the different stages of CRC development and progression and the therapy of this tumor are discussed [14,19,20]. Along with apoptosis and inflammation, autophagy is one of three important mechanisms in CRC. There is a network of complex cross-talk between autophagy and apoptosis pathways in CRC. In human cells, the protein linking the processes of apoptosis and autophagy is beclin 1 (BECN1), essential for autophagosome formation. It interacts with the anti-apoptotic protein Bcl-2 [10]. The interaction of autophagy proteins, which also include BECN1, L-chain microtubule-associated protein 3 (LC3/LC3-II) and autophagy-related (ATG) protein 5 (ATG5) with inflammatory bowel disease (IBD) susceptibility genes is discussed. As is well known, persistent inflammatory bowel disease promotes the development of CRC [10,21]. The high expression of certain ATG proteins (e.g., BECN1, LC3/LC3B-II, ATG5 and ATG6) is associated with a more aggressive CRC phenotype [14]. Autophagy is involved in cell proliferation, survival and metastasis, metabolic dysfunction in the tumor microenvironment (TME), regulation of the immune checkpoints and interaction of microbiota and CRC [19,20,22]. A cross-talk between the DNA damage response and autophagy at the cellular level in CRC is reviewed [23]. The correlation between the process of autophagy and tumor immune infiltration in CRC is also determined [24]. Basic therapeutic trends using autophagy as a new therapeutic target are described, especially in chemotherapy-resistant CRC cells [25].

The role of IGF signaling in the autophagy process in CRC is only partially understood [26,27]. Hence, the aim of the review was to summarize the latest knowledge on the role of the IGF system in the molecular mechanisms of autophagy in the normal colon mucosa and in CRC, taking into account the cellular heterogeneity of the colonic and rectal epithelium.

## 2. Autophagy in Colorectal Cancer—Clinical Implications

CRC, along with other cancers with a mutation of the *Ras* oncogene, is among the “autophagy-dependent” cancers, which are characterized by particularly enhanced autophagy. Many cancer cell lines with activating mutations in *Ras* are observed to have high levels of basal autophagy, necessary for cell growth. The normal growth of some of these cells (including CRC cells) depends on autophagy. Meanwhile, the authors suggest that cells with high levels of mutations in *Ras* could also be more sensitive to agents that inhibit autophagy and simultaneously suppress tumor growth. Additionally, this study also points to a mammalian target of rapamycin (mTOR) signaling-independent mechanism of high basal autophagy induced by *Ras* [28].

Although autophagy is active in the normal colon [29], increased autophagy is usually found in CRC [30,31] both in vivo and in vitro, and will be discussed in more detail later. Here, it can only be mentioned that active autophagy has been shown to contribute to the survival of tumor CRC cells under stress conditions. The tissue expression of the LC3 protein is confirmed in adenocarcinoma cells, but not in surrounding non-cancerous mucosal epithelial cells. The subcellular distribution of LC3 is similar to that in autophagosomes, in cultured cells [30]. It has also been shown that higher BECN1 expression is associated with longer survival of CRC patients, and, together with selected clinical parameters (T stage), may be an independent tumor prognostic factor [31]. Current therapeutic strategies for CRC, based on mechanisms of enhanced autophagy and against the background of broad CRC risk factors, are presented. New perspectives on the use of knowledge on the role of autophagy and unfolded protein response (UPR) as two important mechanisms at the cellular level in response to colon carcinogenesis stress factors are indicated [32]. Additionally, in the course of colorectal carcinogenesis, dual effects of the function of this system, i.e., a system that both inhibits and promotes the survival and proliferation of epithelial cells in colitis-associated cancer are described [33,34]. An oncosuppressive role for autophagy is demonstrated in the early stages of CRC, while in progression from adenomas to carcinomas, autophagy appears to play a tumor-promoting role [34,35]. It has also been proven that the inhibition of mitochondrial autophagy (mitophagy) inhibits tumor growth in both sporadic and colitis-associated cancer models [36].

Autophagy as a mechanism of tumor suppression may also be mediated by genetic changes in proteins involved in the autophagy pathway that lead to increased tumorigenesis [32,37]. Thus, single nucleotide polymorphisms (SNPs) of ATG genes, which potentially influence the occurrence and development of IBD (mainly Crohn’s disease, CD) as risk factors for CRC, may contribute to the development of CRC. In their review, Fritz et al. demonstrated that genes related to autophagy, i.e., *ATG16L1*, immunity-related GTPase M (*IRGM*), leucin-rich repeat kinase 2 (*LRRK2*), intracellular bacterial sensing, nucleotide-binding oligomerization domain-containing protein 2 (*NOD2*) and endoplasmic reticulum (ER) stress, i.e., X-box binding protein 1 (*XBP1*) and orosomucoid like-3 *(ORMDL3*) are important in the pathogenesis of CD. It highlights the important role for microbiome–host interactions in the pathogenesis of this disease [37]. Mechanisms of autophagy regulating the immune response in IBD are also discussed. The genes involved in autophagy/selective autophagy, including *ATG16L1*, *IRGM*, *LRRK2*, *ATG7*, *p62*, optineurin (*OPTN*) and transcription factor EB (*TFEB*) in colitis and in maintaining intestinal homeostasis, are noted [38]. ATG gene mutations are described primarily in types of microsatellite instable-high (MSI-H) and *KRAS*-WT colorectal cancer. Impaired endocytosis and autophagy may provide an alternative pathway for activation of RAS/ERK signaling, resulting in EGFR-dependent tumors [39]. Kang et al. described mutations of several genes in CRC and gastric cancer, namely *ATG2B*, *ATG5* and *ATG9B*, and these only affected cancers with MSI-H. MSI-H CRC showed one or more ATG gene mutations in nearly 28% of patients. These studies suggested that mutations in ATG genes may contribute to cancer development by deregulating the autophagy process [40]. Others have identified changes in the expression and/or epigenetic regulation of ATG genes, such as: *PINK1* (involved in mitophagy) [33], *BECN1* (involved in macroautophagy) and *LAMP2* (involved in chaperone-dependent autophagy) depending on the stage of CRC [41]. This recent study showed results on the role of miRNAs regulating three of the ATG genes mentioned above, and these are *BECN1*-49 mRNA, *LAMP2*-62 mRNA and *PINK1*-6 mRNA. Promoter regions containing at least one CpG region have been detected and described [41].

The potential impact of genetic alterations in ATG genes on the genomic stability and regulation of immune responses in CRC is also highlighted. The CD risk allele Thr300Ala in *ATG16L1* has been shown to be a factor in improved overall survival (OS), with reduced metastasis combined with increased type I interferon (IFN-I) activity in stage I colorectal adenocarcinoma. Thr300A is found to alter mitochondrial anti-viral signaling-dependent IFN-I production in CRC cells [42].

The reduced expression of selected ATG genes/proteins, e.g., ATG5 protein [17], *ATG2B* [43] in CRC vs. control tissue, is shown to correlate with lymphovascular invasion [17] or the infiltration of CD8+ and CD4+ T cells, B cells and T cell receptor signaling pathways [43]. A number of ATG genes can serve as prognostic factors, such as SNP variant rs17094017 A > T in *ATG2B* (better prognosis for patients receiving oxaliplatin-based chemotherapy, increased disease control rate after treatment) [43], *Beclin-1* and *Rab-7* for OS and progression-free survival (PFS) in CRC [44] and *BAX* and poly [ADP-ribose] polymerase 1 (*PARP1*) for CRC recurrence (71.1%) [45]. Other researchers described missense, amplification and deep detection mutations in twenty genes, among them five ATG genes (e.g., *DAPK1*, *ULK1*, *PELP1*, *TSC1* and *CASP3*) that have a mutation rate ≥ 3 and are associated with OS in CRC patients. In addition, eight of these genes (including *SERPINA1*, *DAPK1*, *MAP1LC3C*, *MAPK9*, *TSC1*, *ULK3*, *CASP3* and WIPI1), are found to be prognostically optimal [46]. Another study has added even more valuable ATG genes for prognosis in CRC (e.g., *WIPI2* and *RAB7A*) [47]. Another study listed 11 ATG genes for prediction in CRC [48]. There are genes that differ from those listed in earlier work. The expression of some of these genes (e.g., ephrin receptor B2 (*EPHB2*), interleukin-13 (*IL-13*), microtubule-associated protein 2 (*MAP2*), ribophorin 2 (*RPN2*) and tumor necrosis factor receptor associated factor 5 (*TRAF5*) is positively correlated with MSI. The expression of two of these ATG genes, i.e., *RPN2* and *TRAF5* and additionally *IL-13* correlate with the mutation burden of the tumor [48].

Autophagy also plays an important role in the elimination of microorganisms colonizing the colonic mucosa. A negative correlation has been shown between BECN1 expression and the amount of *Fusobacterium nucleatum* (*F. nucleatum*) DNA in CRC tissue [49]. In regulating CRC metastasis, *F. nucleatum* acts through the caspase activation and recruitment domain 3 (CARD3) and autophagy [50]. The high expression of this bacterium seen in the tissues of patients with CRC recurrence after chemotherapy is also correlated with shorter recurrence-free survival (RFS). An important role of this bacterium in the control of CRC chemo-resistance is also confirmed through the molecular organization of the toll-like receptor 4/myeloid-differentiation primary response 88 (TLR4/MYD88), unc-51 like autophagy activating kinase (ULK1)/ATG7 networks and microRNAs (miR18a* and miR4802) [51]. The involvement of *F. nucleatum* in colorectal carcinogenesis and CRC progression is increasingly highlighted in the review literature. The effects of *F. nucleatum* infection on CRC progression include increased cell proliferation via TLR4 and MYD88 signaling and upregulation of miR21 expression. The activations of TLR4/MYD88 and ULK1/ATG7 signals, which decrease the expression of miR18* and miR4802, respectively, are among the proven mechanisms of action of this bacterium, important both in enhancing CRC cell proliferation, mechanisms of CRC chemo-resistance (5-fluorouracil (5-FU) and oxaliplatin) and in cell autophagy. The beneficial effect of antibiotics targeting *F. nucleatum* in suppressing the growth and progression of CRC by inhibiting pathological mechanisms is taken into account [52].

Among the bacterial species important for CRC growth is *Escherichia coli* (*E. coli*). It is detected more frequently in the colon mucosa of patients with CRC than without cancer. An association between colon infection with the *E. coli* product, colibactin genotoxin (CoPEC or colibactin-producing *E. coli*), and the promotion of autophagy has also been demonstrated [53]. Using a mouse model associated with chronic inflammation (dextran sulphate treatment, DSS), colibactin-producing *E. coli* was confirmed to induce colon carcinogenesis. A role for autophagy in inhibiting these carcinogenic properties of CoPEC is also indicated. At the same time, this is the pioneering study indicating a link between autophagy, CoPEC and CRC [54].

## 3. IGF Signaling and Colorectal Cancer—A Brief Summary

IGF-1 and IGF-2 exert their effects on the cell through specific binding to various membrane receptors (e.g., IGF-1R, IGF-2R, IR and hybrid receptor: IGF-1R/IR) [55]. Most of the actions of both IGFs are mediated by IGF-1R. In addition to mediating the mitogenic and anti-apoptotic actions of IGFs, this receptor can independently affect cell transformation. Of the six IGF BPs (IGF BP1-6), the most commonly manifested circulating form is IGF BP-3, which binds more than 95% of IGFs. The IGF system also includes IGF BP proteases [56,57,58]. A high IGF-1/IGF BP-3 ratio in serum is associated with an increased risk of CRC [3,4,5]. Higher IGF-1 expression is observed in primary CRC cell lines vs. metastatic cell lines [59]. A summary of research on the role of the IGF system in the development of the most common human cancers, including CRC, has already been presented in an earlier review paper [5].

IGF-1R signaling through PI3K/AKT/mTOR complex 1 (mTORC1) has an important and well-documented role in enhancing cell growth, oncogenic transformation and tumor progression. This pathway is associated with cell proliferation, survival and differentiation [4,60]. The deregulation of PI3K/AKT/mTOR signaling is responsible for tumor initiation, chemo-resistance and poor prognosis in CRC [2,5]. IGF-1R signaling affects also the properties of pluripotent stem cells (PSCs) regulating their proliferation and survival [60]. It is postulated that certain CRC subtypes are under autocrine regulation of IGF-1/IGF-1R signaling. Higher levels of IGF-1R gene expression have been demonstrated in tissues with adenomatous polyps and CRC lesions compared to normal mucosa [61,62]. IGF-1R expression levels correlated positively with venous invasion and liver metastasis [61], as well as with tumor size [63]. The focal membranous expression of IGF-1R in Dukes’ C CRC can predict the risk of recurrence, especially liver metastasis [64]. This signaling pathway was also thought to enhance lymphangiogenesis and play a role in metastasis via the lymphatic system in CRC [65]. Other clinical correlations have been shown for rectal cancer, in which a worse response to radiation therapy was observed in patients with IGF-1R overexpression [66]. In vivo and in vitro studies by Codony-Servat et al. have revealed an important role for the nuclear localization of IGF-1R in CRC metastasis. The mechanisms of such protein sequestration involving signal transducer and activator of transcription 3 (STAT3, PIAS3) were explored in vitro. In addition, they highlighted the role of the nuclear localization of IGF-1R as a poor predictor of OS in patients with mCRC [67]. Other attempts have been made to elucidate the mechanisms of IGF-1R action on the increase in neoplastic cell transformation (enhancement of cell motility/migration/invasion) [68,69]. One such factor is the c-Src proto-oncogene [68], and another is the tyrosine-protein kinase Met (c-Met) and urokinase-type plasminogen activator/receptor (uPA/uPAR) system [69]. Evidence for the stimulatory effect of IGF-1R on CRC cell proliferation may also be provided by in vitro and in vivo studies using anti-IGF-1R mAbs (alone or in combination with the chemotherapeutic agent oxaliplatin), which inhibited this process, as well as angiogenesis, increasing apoptosis of tumor cells [70]. A recent multicenter study in IBD patients showed a reduction in serum IGF-1 levels in the prolonged treatment of adult CD patients with adalimumab (anti-TNF-α mAbs). The role of not only GH, but also the impact of ghrelin on GH secretion, and the effect of TNF-α blockade on GH and IGF-1 production in CD treatment with these antibodies were discussed [71].

IGF-1, a conserved, secreted 70-amino acid (AA) peptide, is a critical mediator of many of the biological effects of growth hormone (GH) as a master regulator of its secretion (negative feedback loop) [72,73,74]. IGF-1 binds more strongly to IGF-1R than to IR in its physiological action. Binding to the α subunit of IGF-1R leads to phosphorylation of tyrosine residues and activation of tyrosine kinase. This leads to phosphorylation of adaptor proteins, e.g., IRS-1/2, Shc, 14-3-3, resulting in the activation of a cascade of responses by numerous kinases from intracellular signaling pathways, ultimately modulating the expression of genes associated with apoptosis, autophagy or cell proliferation [55,75,76] (Figure 1).

The overexpression of phosphorylated AKT leads to increased cell division and suppression of apoptosis in 70% of CRC patients, along with the abnormal expression of phosphatase and tensin homolog deleted on chromosome ten (PTEN). AKT also directs signals to the mTOR protein, which promotes angiogenesis and tumor growth [26,75,77]. When activated, mTORC1 regulates not only cell growth and protein synthesis, but also the process of autophagy via ribosomal protein S6 kinase (S6K, RPS6K) and eukaryotic translation initiation factor-binding protein 4E (4EBP1, EIF4EBP1). In contrast, mTORC2 effectors are mainly members of the AGC kinase family, including AKT, protein kinase C (PKC), serum- and glucocorticoid-induced kinases (SGKs). They modulate cytoskeleton organization, cell survival, lipid homeostasis and metabolism [60,78]. It has also been shown that mTORC2 has a key role in mediating TGF-β/activin signaling by directly affecting the activin-induced differentiation of definitive endoderm human embryonic stem cells (hESCs) [79]. Yao et al. demonstrated a role for the IGF-1R/STAT3 pathway in CRC. Phosphorylation of STAT3 resulted in the activation of the transcription factor NANOG, which is responsible for regulating the properties of the epithelial-mesenchymal transition (EMT), on the one hand, and maintaining the stemness of cancer stem cells (CSCs) on the other. In conclusion, the NANOG/Slug pathway in combination with IGF-1R signaling plays an important role in CRC progression [80]. Studies in CRC patients with diabetes mellitus (DM) (*n* = 125) showed a positive correlation between IGF-1, IGF-1R and IR expressions in all cancer tissues. The association of IGF-1 and IR expression appeared to be more pronounced (higher correlation coefficient) in the diabetic group than in the non-diabetic group. According to the authors, this study may suggest a pathogenetic role for IGF-1R/IR signaling in CRC patients with DM [81].

The IGF-2 protein is considered one of three candidates as a selective marker of the progression and staging of CRC. This 7.5-kDa peptide, produced by the liver and many other tissues, is the first gene discovered that undergoes parental imprinting. Loss of imprinting (LOI) or abnormal IGF-2 imprinting can lead to IGF-2 overexpression, increased cell proliferation and CRC development [82].

## 4. Role of IGF Signaling in Regulation of Autophagy

Autophagy is an evolutionarily conserved process in all eukaryotic cells, from yeast to humans, and can be modulated at several stages [83]. The exact mechanisms of this process have been understood relatively recently, resulting in a Nobel Prize for Japanese cell biologist Yoshinori Ohsumi in 2016 [84]. Several forms of autophagy have been described, of which macroautophagy (autophagy) is the most extensively studied. Various studies have demonstrated the selective autophagic degradation of cellular organelles. The best-described type of selective autophagy is autophagy of mitochondria (mitophagy) [12]. For non-selective autophagy, autophagy receptors and adaptors do not appear to be required. Depending on the ways in which the cell contents are incorporated into the lysosome, autophagy is divided into the aforementioned macroautophagy, microautophagy and chaperone-induced autophagy [10,11]. In addition, canonical and non-canonical autophagy are distinguished. The former requires the presence of a complex of about 15 *ATG* products and associated proteins (e.g., DFCP1, AMBRA1, class III PI3K and p150/VPS15 serine/threonine kinase) and represents a process from the formation of an autophagosomal membrane involving the ULK1/ATG13 complex to the formation of ready-made autophagosomes that fuse with lysosomes, which allows the degradation of internally engulfed proteins, lipids and organelles [16,60]. Non-canonical autophagy is an alternative process in which autophagy occurs in the absence of key factors [85,86].

Autophagy is inhibited in most cell types under favorable growth conditions. In contrast, the induction of autophagy occurs in response to starvation, indicating a key role for this process in the adaptation to nutritional conditions. The main negative regulator of autophagy is PI3K/AKT/mTOR signaling, and potential stimulators in normal and cancer cells include sirtuin 1 (SIRT1) and 5′ adenosine-monophosphate-activated protein kinase (AMPK) [11,87,88]. The liver kinase B1 (LKB1)/AMPK pathway can induce autophagy through the phosphorylation (Thr198) and stabilization of cyclin-dependent kinase inhibitor 1B (p27kip1). The stabilization of p27 allows cells to survive growth factor deficiency and metabolic stress through autophagy [89]. In starvation-induced autophagy, there is the inhibition of mTORC1 by AMPK and by the deletion of growth factors (e.g., insulin, IGF-1) [11]. In contrast, under the influence of growth factors, AKT kinase becomes catalytically active and is an important player in regulating mTORC1 activity. In addition, growth factors activate Ras, which stimulates a cascade involving Raf-1, MEK1/2 and ERK1/2. Both AKT and ERK1/2 can phosphorylate one of the two subunits of tuberous sclerosis complex 1/2 (TSC1/TSC2), and AKT can phosphorylate mTORC1 [11,87]. Moreover, AAs activate mTORC1 independently of the AKT/TSC1/TSC2 axis through the GTPase Rag family, which directly interacts with the regulatory-associated protein of mTOR (Raptor) and recruits mTORC1 to the lysosomal surface [11,90].

Studies in the nematodes *Caenorhabditis elegans* indicated that physiological levels of autophagy are a prerequisite for normal cell size. In contrast, abnormal autophagy levels (increased, decreased) result in delayed cell growth. Showing a reduction in cell size for the mutational inactivation of *unc-51*/*Atg1* and *bec-1*/*Atg6*, the authors discussed the mechanisms of this phenomenon. They concluded that insulin/IGF-1/IGF-1R and TGF-β signaling and the certain Atg genes overlapping with these pathways are important in the control of cell size. Autophagy would be expected to act as a key regulatory mechanism for cell growth as well [91].

In HeLa cells and mouse embryonic fibroblasts (MEFs), the disruption of IGF-1R signaling has been shown to reduce autophagy. The inhibition of IGF-1 impairs autophagosome formation. IGF-1R depletion inhibits mTORC2, which in turn reduces protein kinase C (PKCα/β) activity. This disrupts the dynamics of the actin cytoskeleton and reduces the rate of clathrin-dependent endocytosis, which affects the formation of autophagosome precursors. The pharmacological inhibition of the IGF-1R signaling cascade also reduces autophagy in zebrafish and mouse models. These studies support the concept that mTORC2 activity is required to sustain autophagosome biogenesis [92].

Autophagy also controls cellular aging by eliminating damaged cellular components and is negatively regulated by mTOR-mediated IGF/AKT signaling. As mentioned, potential stimulators of autophagy in normal and cancer cells include SIRT1, which affects the regulation of the mTOR pathway [88]. The involvement of other sirtuin family proteins in autophagy has also been studied. In a model of human bronchial epithelial cells (HBECs), it was confirmed that SIRT6 overexpression also induced autophagy by attenuating IGF/AKT/mTOR signaling. Conversely, *SIRT6* knockdown and overexpression of a *SIRT6* mutant (H133Y) inhibited autophagy [93].

Although IGF-1 signaling has been shown to enhance cell growth and exert a cytoprotective effect under acute conditions, it has also been indicated that decreased IGF-1 signaling increases the lifespan in many organisms. In a study by Bitto et al. on WI-38 human diploid fibroblasts, the inhibition of autophagy by IGF-1 signaling, accumulation of cells with dysfunctional mitochondria and reduced long-term cell viability have been observed [94].

The role of IGF-1 in mitophagy has been confirmed. Mitophagy was shown to be promoted only in cells with destructive mitochondrial DNA (mtDNA) mutations when cultured cells were incubated in a serum-free environment. The addition of low levels of IGF-1 prevented this process [95]. Other in vitro studies have shown that IGF-1/PI3K signaling promoted both mitophagy and mitochondrial biogenesis, and the inhibition of IGF-1R activity or expression led to mitochondrial dysfunction. IGF-1 signaling has been found to be essential for maintaining cancer cell viability by stimulating mitochondrial biogenesis and turnover mainly through the induction of transcription activators, namely peroxisome proliferator-activated receptor γ coactivator 1β (PGC-1β) and PGC-1α-related coactivator (PRC), as well as the induction of BCL2/adenovirus E1B 19 kDa protein-interacting protein 3 (BNIP3) accumulation in mitochondria, an important mediator of mitophagy [96].

Studies on MCF-7 (breast cancer), DU145 (prostate cancer) and U2OS (osteosarcoma) cells confirmed the promoting involvement of IGF-1 signaling in autophagy. IGF-1 was shown to mediate mitochondrial-protective signaling coordinated by cytoprotective nuclear respiratory factor-2 (Nrf2). IGF-1 induced expression of the mitophagy receptor BNIP3, through a known AKT-mediated inhibitory phosphorylation on glycogen synthase kinase-3β (GSK-3β), leading to the activation of nuclear factor erythroid 2-related factor 2 (NFE2L2/Nrf2) and acting through downstream transcriptional regulators NRF1 and hypoxia-inducible factor 1 subunit α (HIF-1α). IGF-1 signaling thus couples the induction of mitochondrial biogenesis to basal levels of mitochondrial turnover via Nrf2 and BNIP3, thereby maintaining mitochondrial homeostasis and facilitating tumor progression [97].

Potential mechanisms related to apoptosis and autophagy under hypoxia were also investigated. Hypoxia induced apoptosis, increased ROS production and promoted autophagy compared to normal conditions. IGF-1R coupled to PI3K/AKT/mTOR signaling was shown to play a protective role against oxidative stress and increase cell viability under hypoxia by promoting autophagy and scavenging ROS production. The MEFs with IGF-1R overexpression showed a lower percentage of cells with apoptosis, lower ROS production and higher levels from autophagy as compared to fibroblasts exhibiting IGF-1R disruption [98].

In summary, the main negative regulator of autophagy in physiology is mTORC1 kinase. IGF-1, PI3K and AKT are among the stimulators of mTORC1; therefore, under normal conditions they also inhibit autophagy. A potential stimulator of autophagy under physiological conditions is AMPK signaling. Under pathological conditions (e.g., energy deficiencies, starvation, hypoxia, cancer), IGF-1R signaling promotes autophagy/mitophagy. The overexpression of IGF-1R under hypoxia protects cells from death by altering levels of autophagy and ROS production. However, it may facilitate tumor development and progression (pro-tumoral autophagy role).

## 5. Autophagy in Normal Large Intestine Epithelial Cells—Implication for Colorectal Neoplasia

The epithelial lining of the lumen and the crypts (intestinal glands or crypt units) in the colon is composed of columnar absorptive cells (enterocytes, colonocytes), goblet (oligomucous) cells (GCs), regenerative cells/intestinal stem cells (ISCs) and the less-abundant enteroendocrine cells (EECs) [99,100,101]. In the organization of the colonic crypt, ISCs are located in the lowest parts, giving rise to transit-amplifying (TA) precursor cells, and finally to fully differentiated cells [99]. Given the superficial location of intestinal epithelial cells, it forms the so-called mucosal barrier along with immune cells. With damage to the epithelial barrier, commensal bacteria can move into the subepithelial tissue, inducing the secretion of pro-inflammatory mediators and the recruitment of leukocytes. These phenomena underlie colorectal diseases, such as IBD, ulcerative colitis (UC) and CD [100]. The high heterogeneity of human colonic epithelial cells, together with an understanding of their functions and markers, have become possible by recent molecular techniques at the single-cell level [101,102].

The study showed different proportions of epithelial cells in the small and large intestine. Thus, in the large intestine (colon, rectum), enterocytes accounted for 14% of cells, while GCs in this area were 20% of all cells. By showing the differences in nutrient absorption in the small and large intestine, the authors observed the existence of Paneth-like cells (PLCs) in the large intestine [102]. There were also previously unknown small cellular subpopulations of the ileal epithelium (about 1% of all cells), i.e., bestrophin 4-positive (BEST4+) enterocytes of major importance in the pathogenesis of IBD and CRC [103,104]. These cells demonstrated complex interactions with other cell subsets (e.g., inflammatory fibroblasts and lymphocytes) in UC patients [104].

In a mouse model, ISCs markers in the small and large intestine have been shown to be primarily Lgr5 (also known as Gpr49), which was selected from a panel of intestinal Wnt target genes for its restricted crypt expression. The authors suggested that the stem cell marker they have demonstrated may apply to many normal tissues of the mature organism as well as to cancers [105]. Lying at the base of the crypts are undifferentiated intestinal columnar in shape cells that are considered true ISCs, giving rise to all intestinal epithelial cell lines [99,105,106]. In humans, the replacement of the entire intestinal epithelium takes between 3 and 5 days; hence, the process has to be strictly regulated. Two models for the programming of intestinal crypt cells in the process of regulation and differentiation into mature colonocytes have been described. According to the “pedigree” model, crypt cells are pre-programmed and receive minimal stimuli from the environment, while the “niche” model assumes the influence of the local microenvironment along the crypt on this process. Any dysregulation of normal intestinal epithelial renewal may result in the development of inflammatory changes in IBD, adenoma and CRC [106].

A large number of GCs produce a dense mucus layer that covers the epithelium. The mucus layer covering the human large intestine consists of an inner layer that is normally impermeable to bacteria and a permeable outer layer [107,108,109].

The involvement of most IECs in the mucosal immune system through regulation of the autophagy process has been proven on various research models [100,110,111]. The role of autophagy in the control of the intestinal barrier, metabolism, proliferation and regeneration of both secretory cells and ISCs has been reviewed [112]. IECs were shown to contain autophagosomes during tumorigenesis, and the expression of key ATG genes was induced throughout the process of carcinogenesis. The inhibition of autophagy by the deletion of colonic epithelial cell-specific *ATG7* altered both the initiation and progression of carcinogenesis driven by the loss of adenomatous polyposis coli (*APC*). At the tumor initiation stage, the inhibition of autophagy in IECs led to altered composition of the intestinal microbiota, increased intestinal permeability and activation of the immune response (infiltration of CD8+ T cells) [113].

Autophagy also has a role in the process of intestinal enterocyte differentiation and is associated with the ER stress-related UPR. Terminally differentiated colon epithelial cells were shown to have increased levels of cytosolic Ca^2+^ and activation of all three UPR pathways: inositol enzyme 1 (IRE1), ER-like RNA protein kinase and activating transcription factor 6 (ATF6) compared to undifferentiated cells. The enhanced UPR in differentiated cells was accompanied by the induction of autophagy, as evidenced by an increased LC3 II/I ratio, an increase in Beclin-1 and a decrease in p62 [114].

Autophagy has been shown to play a protective role against UC-like colitis by maintaining normal intestinal microflora and mucus secretion [115]. In addition, autophagy was found to protect against intestinal damage in a DSS-induced epithelial damage model by maintaining the epithelial barrier function and promoting the survival and proliferation of IECs via enhancing STAT3/ERK signaling activation. These results, on the one hand, reveal an important role of autophagy in activating protective, regenerative processes, and on the other hand, may also result in the promotion of colitis-associated cancer [34]. Only one study proved the lack of connection between autophagy in the intestinal epithelium and intestinal tumors, observing only a weak induction of autophagy in the intestinal polyp regions of the mice vs. non-polyp areas [116].

### 5.1. Colonocytes

Colonocytes (absorptive cells, principal cells) differentiate in the basal third of the crypt. Lying on the surface and uppermost crypt, they extend into the lower crypt in the distal colon [117]. As a result of rapid proliferation, they move from the lower parts of the crypts towards the lumen of the colon at a speed of ~1 cell position per hour, after which they undergo exfoliation into the crypt lumen [118]. The total proliferation rate of colonocytes (3–10 billion/day) makes the colon mucosa the organ with the highest proliferation rate of all organs in mammals [119]. Histological studies indicated a species difference in the fate of aging colonocytes. In rats, there is significant luminal exfoliation of aging colonocytes, while in humans, the main pathway for the disposal of colonocytes is mucosal phagocytosis. In this process, aged colonocytes undergo apoptosis and are then phagocytosed by macrophages in the lamina propria or by neighboring epithelial cells [120]. However, Barkla and Gibson do not confirm the luminal shedding in the human colon under normal physiological conditions, supporting the model of “recycling” of senescent cells on the surface of the colonic mucosa [121].

The main function of colonocytes is to absorb Na+, Cl− and water, while secreting K+ and HCO3−. Na+ absorption occurs throughout the human colon due to the presence of Na+ channels located mainly in the apical membrane of surface colonocytes [122]. It has also described the role of a number of K channels distributed mainly in the basolateral membrane of the human colonic crypt in both the physiology and pathology (e.g., IBD) of the large intestine [123]. KCa3.1 channels are thought to be the dominant basolateral potassium channel in human colonocytes, being distributed along the surface–crypt axis [124]. In contrast, activation of the KCa1.1 channel plays a key role in the mechanisms of cell volume regulation and cell death in superficial rat colonocytes [125]. On the other hand, in human colonocytes, KCa1.1 channels are located in the apical membrane of surface cells and the upper part of the crypt, while in UC patients these channels are extended along the entire axis of the crypt [126]. Due to the exposure of the entire colonic epithelium to osmotic stress in physiology, the activation of cell volume regulation mechanisms is required in colonocytes [125]. Key components involved in multilevel transepithelial ion transport in colonocytes (e.g., cystic fibrosis transmembrane regulator (CFTR), epithelial sodium channel, Na-K-Cl cotransporters (NKCC1 and 2), Na-H exchangers (NHE1-4) and colonic H, KATPase, etc.) were presented in one of the most recent excellent reviews [127].

Relatively recently, previously unknown absorptive cells were identified at the top of the colon crypt with the expression of otopetrin proton channel 2 (OTOP2) and satiety peptide, uroguanylin (encoded by *GUCA2B*), able to sense pH, suggesting that these cells have a role in setting the colonic epithelial cyclic guanosine monophosphate (cGMP) tone in response to luminal pH [103]. Markers of these cells, such as mature colonocytes, also include the calcium-sensitive chloride channel, BEST4, which can export bicarbonate; carbonic anhydrase VII (CA7), which catalyzes bicarbonate formation; and the protease cathepsin E. These proteins are involved in electrolyte transport, proton conduction and sense pH [103,104].

#### 5.1.1. Autophagy in Colonocytes

The process of autophagy and the energy metabolism of colonocytes are regulated by the microbiome. Colonocytes use butyrate produced by bacteria as a primary energy source [128]. Colonocyte metabolism closely resembles the metabolic programming of M2 or M1 macrophages. During homeostasis, oxidative phosphorylation dominates, resulting in high oxygen consumption by the epithelium and helping to maintain a microbiome dominated by obligate anaerobic bacteria. Conditions that alter the colonic epithelial metabolism by increasing epithelial oxygenation drive the expansion of facultative anaerobic bacteria, a hallmark of colonic dysbiosis [129].

In the etiology of CRC, especially *F. nucleatum*, enterotoxigenic *Bacteroides fragilis*, *E. coli* and butyrate-producing bacteria are considered [130]. Some of these bacteria may contribute to CRC progression by helping cancer cells evade the immune response by suppressing immune cell function, creating a pro-inflammatory environment or affecting the autophagy process [131]. Mouse colonocytes that had a single encounter with the bacterium showed reduced levels of ATG gene transcripts. Invasive HtrA-positive *E. coli* suppressed cellular autophagy, causing overproduction of free radicals, leading to epithelial hyperproliferation and tumor initiation. After prolonged repeated bacterial invasion, the host epithelium showed an increase in autophagy to eliminate intracellular microbes, and this resulted in the termination of bacterial-dependent epithelial hyperproliferation. Invasive *E. coli* would be expected to contribute to CRC initiation at a crucial time by balancing autophagy and oxidative stress in the colonic epithelium, but they did not act on late-stage tumor growth [132].

Cell-specific functions of autophagy in the colonic epithelial cells are summarized in Table 1.

#### 5.1.2. Colonocytes and IGF-1 System

Normal colonocytes in the rats showed altered levels of IGF-1R (mRNA, protein) and IGF-2R (mRNA) depending on the source and amount of dietary fat, suggesting a potential effect of this food ingredient on the endocrine regulation of colonic cell mitogenesis [133]. Further research is needed to determine the role of obesity (and weight loss) on mitochondrial structure and function, and the effects of altered adiposity on mitochondrial structure and mitophagy in human colonocytes [134].

Studies in normal human colonocytes and in three-dimensional intestinal organoids have uncovered a mechanism for increased cellular DNA damage caused by the direct action of GH via GH-R rather than via IGF-1. This damage may play a role in tumorigenesis [135]. In contrast, in experimental DSS-induced colitis, IGF-1 was shown to contribute to mucosal repair by promoting colonocyte and GC regeneration via β-arrestin2-mediated ERK signaling [136]. In another study, it was observed that an increase in IGF-1 levels after the intravenous infusion of mesenchymal stem cells (MSCs) derived from human ESCs also alleviated colitis in mice by maintaining epithelial cell integrity, repair and regeneration [137]. Similarly, the beneficial effects of IGF-1 on biochemical (e.g., glucose, albumin and total protein levels) and clinical (e.g., maintenance of normal colon size, maintenance of normal body weight and lean mass) and histological (restoration of the mucosal barrier) parameters were demonstrated in experimental DSS-induced colitis in rats. These changes were explained by the re-sensitization of the IGF-1/IRS-1/AKT pathway to exogenous IGF-1 [71].

Using a mouse model with IEC-specific IGF-1-knockout (*IGF-1* KO), it was shown that IGF-1 can increase nutrient uptake, reduce protein catabolism and energy consumption and promote proliferation and expansion of other intestinal epithelial cells. This IEC-specific IGF-1 was responsible for resistance to irradiation and promoted epithelial regeneration. In addition, IGF-1 deletion in IECs increased bacterial translocation to the mesenteric lymph nodes and liver. This IEC-specific IGF-1 loss affected the composition of the gut microbiome. The study indicated an important role for IEC-specific IGF-1 in in vivo intestinal homeostasis, epithelial regeneration and resistance to irradiation, expanding the spectrum of IGF-1 action [138].

Cell-specific functions of IGF-1 signaling in the colonic epithelial cells are summarized in Table 2.

In summary, autophagy plays a special role in colonocytes, which can undergo neoplastic transformation as a result of disturbances at the level of the gut microbiome and elimination of intraepithelial bacteria. Dysbiosis in colonocytes results in both the inhibition of autophagy and an increase in epithelial proliferation during the first stages of tumorigenesis, and an increase in autophagy to eliminate bacteria and arrest cell proliferation. IGF signaling exhibits a protective effect in inflammatory damage to colonocytes, but on the other hand, can lead to tumorigenesis by binding to IGF-1R and activating the entire signaling pathway. The exact role of this system in the mechanisms of autophagy/mitophagy in normal colonocytes has not been defined.

### 5.2. Goblet Cells

The goblet cells (GCs) or oligomucous cells, also known as single-cell glands, are a heterogeneous group of cells, comprising, depending on the gene profile produced, so-called canonical and non-canonical GCs. The main role of GCs is mucus production. By contrast, one of the key regulatory mechanisms in mucus production by GCs is the process of autophagy [110,139,140]. In addition, GCs participate in the immune response through non-specific endocytosis and a pathway called the goblet-cell-associated antigen passages (GAPs) [141,142,143]. It is now believed that these cells not only form mucus but are also closely related to the immune system [143,144]. The proportion of GCs to other IECs increases from the duodenum (4%) to distal colon (16%), in proportion to the number of microorganisms residing in these intestinal segments [107,109].

Although the role of mammalian colon mucus (CM) is relatively poorly understood, it is known to play an important role in the pathogenesis of both IBD [103], as well as CRC [145]. The main component of mucus stored in intracellular granules of specialized GCs in healthy individuals is a glycosylated protein, mucin 2 (MUC2). This glycoprotein maintains the normal structure of the inner CM layer, effectively separating the commensal microbiota from the single epithelial cell layer [108,109,146,147]. Various defects in the inner CM layer have been demonstrated in both mice and humans with UC. The deficiency of MUC2 leads to spontaneous colitis and increased susceptibility to the infection of this part of the colon [108,146,147,148]. On the other hand, high MUC2 biosynthesis induces ER stress and apoptosis in GCs during infectious diseases [144] and can be seen in mucinous colon cancer [149]. Since it was not clear how colon GCs survive by producing highly efficient MUC2, which induces ER stress through misfolded MUC2 proteins, a subsequent study on cultured colorectal adenocarcinoma cells (HT29) showed that high MUC2 biosynthesis actually dysregulated the autophagy processes controlled by IL-22 to maintain the intestinal barrier of innate host defense [150].

#### 5.2.1. Autophagy in Goblet Cells

Closely related to autophagy in GCs is a process called antigen passages (GAPs), involving two related pathways, namely endocytosis and exocytosis [151]. GAPs as pathways delivering luminal antigens to underlying *lamina propria* dendritic cells (DCs) in the steady state occur in both the small and large intestine. GAPs were repressed by TLR ligands in a Myd88 protein-dependent manner. The absence of Myd88 in GCs resulted in the activation of DCs, and consequently colitis [141,142].

Autophagy proteins control the accumulation of mucin granules in colonic GCs. It was shown that in these cells, proteins involved in the initiation and elongation of autophagosomes (e.g., LC3) were required for efficient CM production. In addition, the increased formation of LC3-positive intracellular multivesicular vacuoles in GCs was shown to be influenced by NADPH oxidases (found in the membranes of phagosomes). Thus, both autophagy proteins and endosome formation are necessary for the maximum production of ROS derived from NADPH oxidases [152]. Factors regulating autophagy in intestinal GCs include inflammasome as an activator platform for caspases, in particular NOD-like receptor family pyrin domain containing 6 (NLRP6) inflammasome [153]. A pioneering study identified NLRP6 as the main orchestrator of mucin granule exocytosis in the colon [140]. Studies by Wlodarska et al. indicated that the absence of NLRP6 and other components of the inflammasome (e.g., ASCor caspase-1) in GCs resulted in defects in MUC2 production and increased susceptibility to enteric inflammation via a mechanism of impaired autophagy [140]. However, a subsequent study on the involvement of this protein complex in the regulation of the inner mucus layer yielded results contrary to this concept. It was shown that inflammasome deficiency does not affect the inner mucus layer function. Thus, the involvement of the NLRP6 inflammasome does not appear to be important either for CM formation or for its function at the basal level [154].

A protective role of autophagy against UC-like colitis was demonstrated by maintaining normal intestinal microflora and mucus secretion in GCs. Studies in a mouse model with conditional knockout of *Atg7* (*Atg7* cKO) showed a reduction in the percentage of mucus-producing crypts, the entire CM layer, a decrease in the production of mucin-laden granules in GCs and the amount of fecal MUC2 protein. These changes may have led to the exacerbation of colitis [115].

So-called ”sentinel” GCs (senGCs) have been identified in colonic tissues in multiple mouse strains. These cells undergo non-specific endocytosis and respond to TLR2/1, TLR4 and TLR5 ligands, activating the NLRP6 inflammasome following TLR- and MyD88-dependent ROS synthesis. This triggers the calcium ion-dependent complex exocytosis of MUC2 from senGC and generates an intercellular gap junction signal. This signal in turn induces the secretion of MUC2 from neighboring GCs in the upper crypt, resulting in bacterial expulsion. Thus, senGCs, although dying themselves, guard and protect the colon crypt from bacteria that have infiltrated the inner mucus layer [155]. Other studies indicated higher expression levels of *Il-18* and *Nlrp6* (genes associated with senGCs) in the non-canonical GC population. Similarly, *Wfdc2* and *Areg* genes, whose expressions are altered in colitis [103], were also enriched in non-canonical GCs [156].

Recently, a subpopulation of GCs called intercrypt GCs (icGCs), located on the surface epithelium, has been characterized. They produce mucus different from that secreted by GCs residing in the crypts. The disruption of the icGCs leads to colitis in both mice and humans. Changes in both the organization of intestinal mucus and the number of icGCs are observed in patients with the active form and in UC remission. From a morphological point of view, icGCs appear to belong mostly to highly differentiated canonical GCs (cluster 7, high expression of *Slfn4*) and to cells with high expression of mucus production-related genes (e.g., *Muc2*, *Clca1* and *Fcgbp*). However, from a functional point of view, they show a gene profile related to stress response, cell differentiation, apoptosis and protein transport [156].

#### 5.2.2. Goblet Cells and IGF-1 System

An increase in the number of GCs by 76% in colonic epithelium after the IGF-1 infusion was observed in rats with DSS-induced colitis, which may suggest a direct protective action of IGF-1 on the colonic epithelium [157]. A subsequent study has shown that IGF-1 contributed to the regeneration of the GCs via the β-arrestin/ERK pathway in this model of the study [136]. Other studies have shown that the so-called IEC-specific IGF-1 affected the expansion of GCs and ISCs, confirming the involvement of this peptide in promoting intestinal epithelial renewal [138].

In summary, autophagy in normal colonic GCs plays a role primarily in the control of mucin secretion and thus CM formation. Indirectly, it may also be responsible for the protection of the colonic epithelium from pathogens, and, if impaired, for the development of colitis, as a risk factor for CRC (Table 1). IGF-1 signaling is involved in the regeneration of GCs in colitis. The role of IGF signaling in the mechanisms of autophagy in colonic GCs has not been defined (Table 2).

### 5.3. Intestinal Stem Cells (ISCs)

There are controversies in determining whether the full number of cells in a crypt comes from one or multiple ISCs [99,158]. There are about 5–16 ISCs per crypt. These cells are responsible for the renewal of the intestinal epithelium every 3–5 days [106]. They are characterized by tremendous plasticity. After cell divisions, they give rise to proliferating progenitors (about 120–150), not fully differentiated so-called transit-amplifying (TA) cells. TA cells are located above ISCs within the intestinal crypts [99,159] and are differentiated into five main epithelial cell types [160]. The cellular composition of the small and large intestinal epithelium is controlled by ISCs with phenotype leucine-rich repeat-containing G-protein-coupled receptor 5 (LGR5) [101].

The involvement of both SCs and non-SCs, which may dedifferentiate into the SC-like phenotype in the course of intestinal tumorigenesis, has been suggested. The role of both populations of these cells is also discussed in human colon carcinogenesis [159]. One concept of colon carcinogenesis involving ISCs was the so-called “top-down” model, in which tumor initiation would begin at the top of the crypt (only here, cells with *APC* mutation were observed), and it would spread laterally and also downwards towards the normal crypt [161]. Other author proposed the so-called “bottom-up” spread of cancer. In patients with a familial predisposition to *APC* mutation, the presence of dysplastic lesions within whole and then single crypts has been observed; therefore, this direction of the spread of lesions involving ISCs is not excluded [162].

#### 5.3.1. Autophagy in Intestinal Stem Cells

Studies suggested that autophagy regulates the maintenance, expansion and differentiation of SCs [163], including ISCs [164,165]. The study by Groulx et al. pointed to an active autophagy process in normal colonic glands, where it concerned the primarily intestinal proliferative/undifferentiated and progenitor cell population [29]. Asano et al. in a mouse model of the study showed that intrinsic autophagy promoted ISC maintenance by reducing excessive ROS [164].

Particularly important seems to be the role of autophagy in the early stages of cell reprogramming during the generation of induced PSCs [166]. Autophagy has been shown to play an important role in controlling ISCs’ proliferation, which may have clinical implications in CRC [167,168]. A critical gene for inhibiting ISC proliferation and maintaining intestinal epithelial homeostasis is the *Drosophila* sorting nexin 1 (*SH3PX1*). It operates through an extended endocytosis/autophagy network that includes many other genes that similarly affect ISCs (e.g., *shi*/*dynamin*, *rab5*, *rab7*, *atg1*, *atg5*, *atg6*, *atg7*, *atg8a*, *atg9*, *atg12*, *atg16* and *syx17*). In addition, the excessive activation of ERK, calcium signaling and ER stress have been shown to autonomously stimulate the proliferation of ISCs. Increased cell division induces epithelial stress and the activity and production of other proteins (e.g., Yki, Upd3 and Rhomboid) also in enterocytes, catalyzing the conjugated hyperplasia of ISCs [167]. Other studies also confirmed that repression of autophagy dramatically increases EGFR and phosphorylated MAPK/ERK levels. Furthermore, using TCGA data, a strong association was discovered between CRC with somatic mutations in *ATG* and *SNX*, and MSI-H and CpG island methylator phenotype (CIMP). In addition, a negative association was found between mutations in three human nexins sorting genes (*SNX9*, *SNX18* and *SNX33*), *SH3PX1* orthologs in *Drosophila* and activating *KRAS* mutations in CRC. This suggests that in humans, as in flies, autophagy deficiencies may promote CRC through the activation of the EGFR/MAPK/ERK pathway in a mechanism that is independent of activating mutations in *EGFR*, *KRAS* and *BRAF* [39]. It was also shown that an important role in the occurrence and development of UC is Slit2/Robo1 signaling in the mechanism of modulation of the autophagy process in ISCs Lgr5(+) [169].

Using a model of organoids cultured from mouse intestinal crypts and in vivo experiments, the interactions between ISCs and microbiota were studied. Lgr5(+) cells showed the constitutive expression of the NOD2 at a much higher level than in Paneth cells. Stimulation of NOD2 by its agonist, muramyl dipeptide (MDP), a peptidoglycan motif common to all bacteria, triggered the survival of ISCs, leading to strong cytoprotection against oxidative stress-induced cell death [170]. Further studies showed that MDP induced a significant reduction in total and mitochondrial ROS, which is associated with the induction of mitophagy. These findings elucidated a mechanism of NOD2-mediated cytoprotection involving the removal of lethal excess ROS via mitophagy, induced by the coordinated activation of NOD2 and ATG16L1 through the nuclear factor κB (NF-κB)-independent pathway [171].

#### 5.3.2. Intestinal Stem Cells and IGF System

IEC-specific IGF-1 involvement in intestinal epithelial regeneration has been confirmed through its effect in the expansion of ISCs and secretory cells [138]. In a mouse model, mesenchymal IGF-1 signaling was shown to drive crypt regeneration following the SC loss from whole-body 12-Gy γ-irradiation. IGF-1 released from pericytes stimulated mTORC1 in facultative SCs to regenerate lost crypt base columnar cells [172]. Lgr5(+) ISCs maintained the epithelium in the crypt and modulated the proliferation of its cells in response to niche signaling. It was shown in a mouse model that one of the factors stimulating the “acute” entry of Lgr5(+) ISCs into the S-phase of the cell cycle was GLP-2 acting through GLP-2R, and that IGF-1R signaling via IGF-1 was involved in the chronic induction of Lgr(5+) ISCs’ expansion [173].

In summary, autophagy regulates the maintenance, expansion, differentiation and proliferation of ISCs. A role for autophagy in the generation of active pluripotent stem cells is suggested. A link between mitophagy and ROS scavenging and bacterial stimulation of MDPs in ISCs has also been demonstrated, thus linking microbiota and epithelial regeneration. Exploring the mechanisms of autophagy/mitophagy in ISCs may have clinical implications and therapeutic applications (Table 1). IGF-1 signaling is involved in the chronic induction of Lgr(+) SC expansion. Although research on the direct involvement of IGF signaling in autophagy in ISCs is lacking, this pathway is important in maintaining local intestinal homeostasis (Table 2).

### 5.4. Paneth Cells and Paneth-like Cells

Paneth cells (PCs), also known as “niche cells” in the small intestine, serve as multifunctional guardians of ISCs, both by secreting bactericidal products (e.g., lysozyme and α-defensins/cryptdins) and by providing vital signals about the niche [174,175]. PCs have been found to respond to bacterial invasion, i.e., *Salmonella typhimurium* (*S. typhimurium*) by redirecting a critical secreted anti-microbial protein in a process called secretory autophagy. Unlike the fusion of conventional autophagosomes with lysosomes, these lysozyme-containing vesicles are released into the intestinal lumen [175,176].

Sato et al. proposed that Lgr5(+) ISCs compete for available PC space, the number of which is tightly regulated [174]. PCs are formed directly over the base of the crypt, originally referred to as the “stem cell zone” [177] and can also serve as detectors of nutritional status and enhance the function of ISCs in response to calorie restriction [178]. A more recent study, however, indicated that blocked PCs are being replaced by enteroendocrine and tuft cells. These, in turn, occupy the positions of the PCs between the Lgr5(+) ISCs, serving as an alternative source of Notch signals, which are necessary to maintain the Lgr5(+) ISCs [179].

Although one research study has failed to detect PCs in normal human colons [103], another study has described a population of Paneth-like cells (PLCs) in the human adult large intestine [102], previously described as CD24(+) cells [174]. PLCs in the large intestine and PCs in the ileum share a common set of highly expressed genes that include not only microbial-defense genes, such as lysozymes involved in lysosome function and neutrophil activation, but also genes encoding niche factors that support the Lgr5(+) ISCs (e.g., EGF, wingless-related integration site 3 (Wnt3), Notch, ephrin A/B and platelet-derived growth factor (PDGF) ligands) [102,174]. In human colonic PLCs, the expression of N-acetylglucosamine-1-phosphate transferase subunits alpha and beta (GNPTAB) and superoxide dismutase 3 (SOD3) has been specifically demonstrated [102], as well as the chromatin organizer special AT-rich binding protein 2 (SATB2), involved in carcinogenesis (including CRC) [180].

#### 5.4.1. Autophagy in Paneth Cells

As in the case of GCs, autophagy facilitates the secretory functions of Paneth cells [110]. It was shown that PCs deficient in ATG genes (e.g., *ATG16L1* and *ATG5*) exhibited significant abnormalities in the exocytosis of granules of these cells. In *ATG16L1* deficiency, the increased expression of genes involved in peroxisome proliferator-activated receptor (PPAR) signaling and lipid metabolism, acute phase responses and two adipocytokines (e.g., leptin and adiponectin) were observed. It was also found that CD patients carrying the *ATG16L1* risk allele had an increased percentage of PCs with disordered or reduced granules, similar to those observed in autophagy protein-deficient mice, and they were characterized by increased levels of leptin protein [181]. The loss of *Atg7* in the intestinal epithelium led to PC abnormalities (i.e., aberrant granules) similar to those associated with the loss of function of both *Atg16L1* and *Atg5.* The study suggested that a defect in the autophagy pathway in the intestinal epithelium is responsible for PC pathology [182].

Another author has shown in a mouse model study that the Atg7 protein was required to protect the colon epithelium against *Citrobacter rodentium* (*C. rodentium*) infection. In the *Atg7* KO mice model, an increase in the mRNA of pro-inflammatory cytokines in the colon and an enhancement of the clinical symptoms of disease were observed after infection with *C. rodentium* (strain DBS100) [183].

Bel et al. showed that during bacterial invasion by *S. typhimurium*, PCs underwent ER stress leading to secretory autophagy. Bacterial-induced fragmentation of the Golgi apparatus disrupted the normal secretory pathway and activated the ER stress response. The lysozyme was redirected to this alternative secretory pathway based on autophagy. Autophagy required external signals from innate lymphoid cells and restricted bacterial proliferation. Mutation in *Atg16L1* increases risk of CD in humans [175]. The importance of secretory autophagy of lysozymes in PCs are commented on in detail in other articles. Overall, these studies point to the role of a new, alternative system of so-called secretory autophagy in host defense mechanisms, which may have clinical implications in IBD (predisposition to some forms of CD). In this dual mechanism, secretory autophagy (secretion of the lysozyme from secretory vesicles of PCs in response to bacterial intestinal infection) is triggered involving both intracellular (ER stress) and extracellular (subepithelial innate immune cells) factors [176,184].

#### 5.4.2. Paneth-like Cells and IGF System

To date, there is a lack of research on the direct role of IGF signaling in the normal function of PLCs or the interaction between the IGF system and autophagy in these colonic cells. In contrast, IGF-1 has been shown to be an important mediator of the tropic effect of glucagon-like peptide 2 (GLP-2), which attenuated small intestinal inflammation and improved the immune response of intestinal PCs in the total parenteral nutrition animal model of a study [185].

In summary, the involvement of autophagy disorders in PLCs in the pathogenesis of IBD, mainly CD, is proven. The results show that autophagy in PLCs plays an important role in the host defense against bacterial infections (e.g., *C. rodentium* and *S. typhimurium*) and in the regulation of colitis caused by these pathogens (Table 1). The mechanism of secretory autophagy of the lysozyme in PCs, which is regulated by cellular intrinsic (ER stress response) and extrinsic (innate immune system) mechanisms, has been described. There is a lack of papers on the direct involvement of IGF signaling in autophagy in PLCs (Table 2).

## 6. Cells of the Tumor Microenvironment

TME is a complex network of cancer cells and various populations of stromal cells (e.g., monocytes, macrophages, neutrophils, fibroblasts, infiltrating T and B cells and adipocytes), as well as extracellular matrix (ECM) components, soluble factors and signaling molecules [186,187,188].

### 6.1. Epithelial Colorectal Cancer Cells (Epithelial CRC Cells)

A dual role of autophagy, both as a tumor suppressor system and to promote survival and proliferation of epithelial cells in colitis-associated cancer, is proven [33,34]. The mechanisms involved in the induction or inhibition of autophagy under various experimental conditions are only partially understood. Thus, the research hypothesis that cancer cells (including CRC cells) use autophagy to obtain nutrients (e.g., AAs) from digested organelles as alternative energy sources was confirmed quite early. The formation of autophagosomes in numerous CRC cell lines and their rapid consumption under conditions of nutrient starvation have been observed. Inhibitors of autolysosomes and autophagosomes, i.e., 3-methyladenine (3-MA), significantly enhanced apoptosis under conditions of AA and glucose deficiency. Similar results were obtained in cells with a reduced expression of ATG7 [30]. ATG7 deletion in cancer cells has also been shown to induce a stress response characterized by the activation of AMPK and p53 and a decrease in glycolytic enzymes. Blocking autophagy would therefore impede tumor growth by inhibiting cancer cell metabolism [113]. In the azoxymethane (AOM)/DSS-induced CRC model in vitro and in sporadic CRC, the inhibition of tumor growth was also demonstrated by a loss of *ATG7* (exclusion of autophagy). At the same time, an important role of mitophagy in nutrient acquisition in CRC was revealed. The loss of mitochondrial recycling through the inhibition of mitophagy inhibited CRC cell growth [36]. In vitro studies have confirmed the observed in vivo, reduced expression of ATG5 compared to its expression in normal colonic cell lines, suggesting that ATG5 may function as a tumor suppressor [17].

Autophagy also plays an important role in the elimination of microorganisms (mainly *F. nucleatum*) from CRC cells. Some mechanisms responsible for this process have been revealed. It has been shown that the *F. nucleatum* infection increased the motility of CRC cells and elevated the expression of CARD3, LC3-II, BECN1 and vimentin, and decreased the expression of E-cadherin and p62 in tumor cells [50]. In HCT116 and HT29 cells, the pathogen promoted the activation of tumor autophagy, increasing the expression of the phosphorylated AMPK (pAMPK), pULK1, ULK1 and ATG7 [51]. Similarly, the infection of cultured HCT116 cells and susceptible mice with a 11G5 strain of *E. coli* product, the colibactin (CoPEC), induced autophagy and DNA damage repair, while infection with a mutant strain without colibactin (11G5DclbQ strain) did not cause these changes. The loss of *ATG16L1* in CRC cells increased markers of inflammation, DNA damage and cell proliferation, and enhanced carcinogenesis in ApcMin/^+^ mice infected with the 11G5 strain of CoPEC [53].

In different subtypes of CRC cells (e.g., WT p53, mutant *p53* and p53^(−/−)^, APR-017 methylated p53-reactivation and induction of massive apoptosis-1 (PRIMA-1^met^; APR246) were shown to have an inductive effect on autophagy. The increase in autophagy was independent of p53 status. PRIMA-1^met^ was responsible for the formation of autophagic vesicles (AVs), AV fusion with lysosomes and increased lysosomal degradation in a mechanism of upregulation of mTOR/AMPK/ULK1/Vps34 autophagic signaling. In addition, autophagy played a key role in the inhibitory effect of PRIMA-1^met^ in cells bearing WT p53 [189].

In vitro models using various cancer cell lines (including CRC cells) indicated that hypoxia-induced autophagy is part of a general cell survival mechanism that is controlled by HIF-1. It has been shown that the expression of the proapoptotic proteins BNIP3 and BNIP3L is required for optimal induction of autophagy in hypoxia, implicating their atypical Bcl-2 homology 3 (BH3) domains in the autophagy process without inducing cell death. The atypical BH3 domain in *BNIP3* and *BNIP3L* (two HIF target genes) can compete with Beclin 1/Bcl-2 and Beclin 1/Bcl-X(L) complexes, releasing Beclin 1 from the complex and subsequently increasing autophagy [190].

#### IGF Signaling in Autophagy of CRC Cells

The insulin/IGF-1/PI3K/AKT/mTOR signaling under normal conditions is a negative regulator of autophagy. In contrast, under hypoxia, IGF-1 signaling acting through suppression of the “canonical” PI3K/AKT/mTORC1 pathway stimulates autophagy [26,98]. Interestingly, different classes of PI3K control the autophagy pathway in distinct directions. As shown in HT29 cells, the activity of a class III PI3K product was required for autophagic sequestration, while the stimulation of class I PI3K activity inhibited the autophagic sequestration and degradation [191]. It has also been shown how the autophagic lysosomal pathway can be activated by inhibiting the PI3K/AKT/mTOR signaling [192,193]. HT29 cells have been shown to express WT PTEN: a tumor suppressor that negatively regulates IL-13-dependent PI3K/AKT (PKB) signaling. In addition, PTEN (via lipid phosphatase activity) is involved in autophagy, along with AKT/PKB acting downstream. These results indicate a novel function of PI3K/PTEN/PKB signaling, as well as a new link between autophagy control and tumor progression [192]. In contrast, the knockout of AKT or inactivation with small molecule inhibitors markedly increased autophagy. Treatment with a lysosomotropic agent (chloroquine) resulted in the accumulation of abnormal autophagolysosomes and ROS, leading to accelerated cell death in vitro and complete tumor remission in vivo [193].

The role of mTORC2 as a negative regulator of chaperone-mediated autophagy through its ability to transmit signals to various effectors that modulate autophagy in mammals, e.g., AKT, PKC, serum and glucocorticoid-regulated kinase 1 (SGK-1) and forkhead box class O (FOXO) transcription factors is debated [194]. In CRC cells with constitutively activated mTORC1, mTORC2 appeared to regulate basal autophagy levels and participate in the maintenance of signaling vesicles and tumor cell survival. In most of the CRC cell lines analyzed, autophagy was not induced by receptor tyrosine kinase (RTK) inhibition, which was due to constitutive PI3K/AKT/mTORC1 signaling. Interestingly, CRC cells exhibited detectable basal autophagy despite RAS and PI3K activation. Unexpectedly, the inhibition of basal autophagy affected the tyrosine phosphorylation of several RTKs, in particular c-MET [195].

IGF-1 itself regulates cell survival, proliferation, differentiation, apoptosis and metabolism CRC cells, as has been shown in vitro and in vivo [196,197,198]. CRC cell lines isolated from a primary tumor site showed higher IGF-1 expression among other growth factors and proangiogenic factors compared to metastatic cell lines [59]. Several types of TME cells, including CRC cells, also expressed IGF-1R. The administration of IGF-1 to cultured CRC cells (MC-38, HCT116 and DLD-1) resulted in increased cell migration, which was inhibited by the use of the inhibitor of IGF-1R signaling, i.e., NT157. In a mouse model, it has been shown that tumor growth and inflammation of the metastatic niche can be inhibited by dual inhibition of IGF-1R and the STAT3 pathway by the aforementioned NT157, resulting in significantly smaller sizes of both primary and metastatic tumors [199].

As for autophagy, it has been shown to increase cell survival in glucose-deprived environments by increasing autophagy through inhibition of the IGF-1/PI3K/AKT/mTOR signaling [200]. This has been demonstrated in the DLD-1, HT29 and colon26 cells in a glucose-deprived medium, where a reduction in phosphorylated mTOR (pmTOR), IGF-1, pAKT and PI3K, and increase in LC3-1, LC3-2, pAMPKα and PPARα expression were observed, also influenced by adiponectin. In another study on HCT116 and SW480 cells, a significant role for pleckstrin homology-like domain family A member 2 (*PHLDA2*) in autophagy has been demonstrated. The knockout of the imprinted *PHLDA2* blocked its activity. This resulted in the inhibition of cellular proliferation, invasion, migration and EMT in vitro, while it promoted apoptosis, in part through the activation of autophagy. The promotion of autophagy and inhibition of EMT occurred at least in part through PI3K/AKT/mTOR and GSK-3β signaling pathways [201].

IGF-1 has also been shown to inhibit the autophagy process in 5-FU-resistant (HCT-8R5-FU) cells through AKT/mTOR pathway activation [202]. Treating HT29 cells with calycosin (methoxyisoflavone) and after incubation with chloroquine (inhibitor of autophagy), IGF-1 (as an AKT/mTOR activator) or EX-527 (as a SIRT1 inhibitor) completely prevented calycosin-induced autophagy. These studies suggested that cell invasion and division are mainly controlled by autophagy and mediated by SIRT1/AMPK and AKT/mTOR signaling [88].

In summary, under conditions of energy deficiency and bacterial infection of the colonic mucosa, there is an increase in autophagy/mitophagy in CRC cells, which serves to improve the metabolism of tumor cells and remove microorganisms from the cells. The decreased expression of ATG genes (loss of *ATG5* or *ATG7*) or impaired mitochondrial recycling inhibits autophagy, resulting in the inhibition of CRC growth. IGF-1/IGF-1R signaling regulates CRC cell proliferation, invasion, migration, apoptosis and autophagy. The interactions of IGF signaling and autophagy in neoplastic CRC cells are complex (Figure 2). Under hypoxia, there is an increase in autophagy induced by the suppression of PI3K/AKT/mTOR signaling via IGF-1R. On the other hand, the IGF-1-mediated inhibition of autophagy is evident in the 5-FU-resistant CRC cells.

### 6.2. Colorectal Cancer Stem Cells

The current attractive hypothesis of CRC initiation and growth assumes the involvement of cancer stem cells (CSCs): a small subpopulation of CSCs with embryonic stem cell (ESC) characteristics [99,203]. Such cells would also be the driving force behind tumor progression and metastasis. A number of genes involved in their induction and pluripotency markers of CSCs (e.g., *OCT4*, *SOX2*, *NANOG*, *c-Myc* and *KLF4*) have been identified [204]. In a mouse model mimicking the clinical development of human CRC, Lgr5 was shown to identify intestinal CSCs. Interestingly, selective ablation of Lgr5(+) cells reduced the growth of the primary tumor, but did not result in tumor regression, and there was even rapid tumor recurrence by proliferating Lgr5(−) cells that replenished the pool of Lgr5(+) CSCs. CSCs were further shown to be critical for the formation and maintenance of CRC metastasis to the liver [205]. Human LGR5(+) cells in CRCs were confirmed to serve as CSCs in growing tumor tissues [203].

The study by Yao et al. identified a small subpopulation of NANOG(+) CRC cells with features of CSCs and EMT [80]. NANOG is a multipotential transcription factor that helps mouse ESCs maintain pluripotency by suppressing cell determination factors [206]. This factor modulates EMT and metastasis by binding to the Slug promoter, and transcriptionally regulates Slug expression. Moreover, these are pioneering studies revealing that NANOG is regulated by the IGF-1R/STAT3 pathway in CRC [80].

#### 6.2.1. Colorectal Cancer Stem Cells and Autophagy

Experimental studies indicated that CSCs are in a state of autophagy, and the blockade of autophagy reduces their activity and sensitizes them to anti-cancer drugs. The process of mitophagy in CSCs plays a role mainly in the metabolic reprogramming of these cells [207]. Only one study has addressed the mechanisms of autophagy associated with CSCs in CRC. Wu et al. demonstrated a key role of the caudal type homeobox 1 gene (*Cdx1*) in chemo-resistance to anti-cancer agents of the type of microtubule inhibitors (e.g., paclitaxel) through activation of autophagy. *Cdx1* is involved in a complex network of p53 and Bcl-2 signaling. The *Cdx1*-mediated activation of autophagy in colorectal CSCs resulted in the inhibition of cell apoptosis by Cdx1/Bcl-2/LC3 signaling. In turn, p53 plays a major role in apoptosis and inhibits autophagy in the colorectal CSCs [208].

Hypoxia also increases the ability of CSCs (here called tumor-initiating cells, TICs) to self-renew, while inducing proliferation arrest in their more differentiated counterparts. Autophagy has been revealed as one of the major pathways induced by hypoxia in TIC-enriched patient-derived CRC cultures. Hypoxia-induced autophagy stimulated phosphorylation of the ezrin at the Thr567 site. In addition, PRKCA (PKCα) was identified as a potential kinase involved in the hypoxia-induced autophagy and self-renewal of TICs [209].

#### 6.2.2. Colorectal Cancer Stem Cells and IGF System

IGF-1 signaling has been shown to regulate the cancer stemness in various models of SCs, including colorectal cancer SCs. Chemo-resistance to 5-FU (5FU-R) and oxaliplatin (OxR) in CRC cells were shown to have markers and phenotypes of CSCs. HT29/5FU-R and HT29/OxR cells were approximately five-fold more sensitive to IGF-1R inhibition compared to parental cells. Chemotherapy-induced IGF-1R activation provides increased sensitivity to IGF-1R-targeted therapy [210]. The different activation of IGF-1R signaling and tumor formation is also highly correlated with the diversity of niches. The influence of modulation of complex niche parameters (e.g., hypoxia, inflammation and extracellular matrix), as well as target SCs (ESCs, germinal SCs, mesenchymal SCs) and CSCs is emphasized, with reference to tumor reprogramming involving IGF-1R signaling [76].

Increased diet-induced IGF-1 and insulin levels, which are observed in obesity in Western countries, may lead to the increased proliferation and inhibition of apoptosis of CSCs via PI3K/AKT/Wnt signaling. The role of CSCs in diet-induced CRC is debated [211].

It has also been shown that colorectal CSCs with IGF-2 LOI have a higher level of autophagy than CSCs with maintenance of imprinting (MOI). Moreover, IGF-2/IR-A was found to play a more important role in CSCs’ formation than IGF-2/IGF-1R signaling. These results suggest that IGF-2 LOI can promote the pluripotency of CSCs in CRCs by promoting CSCs’ autophagy (Figure 3). After miRNA-195 degradation, IGF-2 exhibited a greater ability to interact with IR-A overexpression than IGF-1R, which could further activate CSC autophagy [212].

In summary, colorectal CSCs undergo constant autophagy. Mitophagy serves to metabolically reprogram these cells. IGF-1R signaling regulates the cancer stemness. The chemo-resistance of CRC cell lines with the phenotype of CSCs is associated with greater sensitivity to inhibition via IGF-1R signaling. Indirectly, this pathway may be involved in modulating EMT and metastasis of CRCs through the regulation of NANOG. IGF-2 LOI promotes pluripotency of CSCs by promoting autophagy.

### 6.3. Cancer-Associated Adipocytes (CAAs)

The hypothesis that adipocytes in TME serve as energy providers and metabolic regulators to promote the growth and survival of CRC cells was confirmed. Using adipocytes isolated from CRC patients, a close interaction was identified between CAAs and CRC cells in promoting mitochondrial fatty acid oxidation and autophagy in CRC cells via AMPK signaling. A co-culture of CRC cells with adipocytes for 48 h, using IGF-1 as a chemoattractant in transvascular migration assays, showed that adipocytes promoted dedifferentiation and increased the aggressiveness of CRC cells [213]. Another study has shown that genes involved in pyruvate, glucose and lipid metabolism, fibrosis and inflammation processes are central in the transcriptional reprogramming of adipocytes found in obese and CRC patients. Exposure of adipocytes to ω3 and ω6 polyunsaturated fatty acids (PUFAs) in vitro modulated the expression of genes involved in processes potentially associated with carcinogenesis. The transcriptional program of adipocytes appears to be strongly influenced by body mass index (BMI) status in CRC patients [214].

However, a relatively recent study on mouse mature adipocytes (3T3-L1 cells) also indicated that autophagy was positively associated with adipocyte differentiation and was regulated through insulin/PI3K/AKT/mTOCR1 signaling. Autophagy may have a direct suppressive effect on insulin signaling through a feedback mechanism that inhibits AKT phosphorylation, thereby promoting the differentiation of mature adipocytes induced by dysfunction in insulin signaling [215]. It should be remembered that IGF-1 belongs to obesity-associated adipokines, proteins that along with leptin, insulin and IL-6 are also produced by adipocytes. This may have a close etiological link to the occurrence of CRC, which is sometimes called obesity-associated cancer [32,216].

In summary, autophagy in CAAs is correlated with adipocyte differentiation and is regulated by insulin/PI3K/AKT/mTOCR1 signaling. Studies on the precise mechanisms of regulation of autophagy involving the IGF pathway in CRC-associated adipocytes are lacking (Figure 3). The dysregulation of complex processes important for adipocyte function may contribute to more favorable conditions for tumor formation or promote tumor progression, thus linking obesity and CRC. The need of the moment is to develop strategies to help prevent obesity (and indirectly CRC) by regulating insulin and insulin/IGF signaling.

### 6.4. Tumor-Infiltrating Immune Cells (TIICs)

The immune cell population within the TME of primary CRC includes mainly tumor-associated macrophages (TAMs), lymphocytes and tumor-associated neutrophils (TANs), with relatively few plasma cells and eosinophils [186,187,217,218].

Considering the high heterogeneity in TME immune cell infiltration in CRC, three subtypes of autophagy were distinguished [219]. In cluster 1, the greatest infiltration of innate and adaptive immune cells, increased stromal activity (including activation of EMT, TGF-β and angiogenesis signaling pathways) and the greatest infiltration of fibroblasts and endothelial cells (ECs) were present. The expression of immune checkpoint molecules was also significantly elevated. Cluster 2 was characterized by less infiltration of immune cells and enhancement of DNA damage repair pathways, while cluster 3 was characterized by immunosuppression. The distinguished types of autophagy had prognostic significance in CRC. It was shown that a high autophagy characterization score (APCS) signature resulted in shorter survival and a significantly positive correlation with TME stromal activity. A low APCS, relevant to activated damage repair pathways, showed an enhanced response to anti-programmed cell death 1/programmed death-ligand 1 (PD-1/PD-L1) immunotherapy. ATG genes were thought to be responsible for the heterogeneity of immune cell infiltrations in TME in CRC [219].

TAMs account for up to 50% of the tumor mass and occur in the form of M1 macrophages, serving a tumor-preventing role, and M2 macrophages with a tumor-promoting activity [220]. An in vitro study (LoVo cells) indicated that the upregulation of autophagy in TAMs inhibited proliferation and promoted apoptosis of CRC cells, improving the efficacy of radiation therapy [221].

In the high-risk autophagy group, there was reduced infiltration of M1 macrophages and increased infiltration of T regulatory cells. From CRC-specific transcriptomes, a prognostic 10-gene ATG signature was determined that effectively predicted disease-free survival (DFS) for patients with early-stage (stage I/II) CRC. It was shown that CRC patients in this group can have tumor recurrence through an anti-immune/anti-inflammatory response [222]. Another analysis of the prognostic value of ATG genes in CRC, considering the relative proportions of TIICs and the mRNA expression of many key immune checkpoint receptors, showed that the autophagy signature was closely related to immune TME and independently of other clinical factors. Based on the 11 ATG genes (including *CX3CL1*, *ULK3*, *CDKN2A*, *NRG1*, *ATG4B*, *GAA*, *RGS19*, *DDIT3*, *GRID1*, *DAPK1* and *SERPINA1*), a signature for predicting CRC prognosis was developed. The proportion of MSI-L and MSI-H CRC in the high-risk group was higher than in the low-risk group. A study by this author indicated that poorer survival outcomes in CRC patients with high-risk scores may be caused by an immunosuppressive microenvironment [223].

Interactions between CRC-associated obesity and inflammation regulated by AT macrophages adjacent to the tumor, which secrete pro-inflammatory cytokines, e.g., TNF-α, monocyte chemoattractant protein-1 (MCP-1) and IL-6 have long been highlighted [224]. A recent study on an in vitro and in vivo mouse model of obesity has confirmed the relationship between obesity, AT-specific IGF-1, macrophage-associated AT inflammation and polarization of TAMs to M2 phenotype, as potential drivers of CRC development in women deprived of ovarian hormones [225].

The effect of TME cell type-dependent autophagy on colitis and colitis-associated cancer (CAC) has also been noticed. For autophagy in TAMs, it has been shown to inhibit colitis, but its function in CAC is unknown. Autophagy in natural killer cells (NKs) inhibits colitis but promotes CAC. With regard to other immune cells within the TME, the data indicate the promotion of both colitis and CAC through autophagy in TANs and fibroblasts. In contrast, autophagy in DCs and T lymphocytes has an inhibitory effect on colitis and CRC. More debatable is the role of autophagy in ECs in both diseases. The effects of autophagy in different TME cells depend also on the stage of colitis-associated CRC [226]. Data on the exact effect of the IGF system components on autophagy in tumor-infiltrating immune cells in CRC are lacking (Figure 3).

### 6.5. Cancer-Associated Fibroblasts (CAFs)

CAFs are a heterogeneous group of cells of unclear cellular origin. In addition, they are the most abundant component of non-cancerous cells in the TME [227,228]. They are involved in both tumor growth and progression and have been attributed significant prognostic significance. CAFs have a dual inhibitory role in the context of immune cells, preventing both access to the TME and proper function within the tumor [227,229,230]. The mechanisms of the autophagy-related process in CAFs, including correlations between pro-autophagy factors (e.g., hypoxia, glycolysis, senescence, anti-tumor chemicals, and cytokines) and autophagy have been described. It was demonstrated that cytokines produced by CAFs promote tumor survival by secretory autophagy. Finally, it has been shown that autophagy in CAFs affects cancer cells, regulating tumor stemness and progression [231]. Oxidative stress in fibroblasts induces mitophagy. Subsequently, the fibroblasts are forced to undergo aerobic glycolysis, and produce energy-rich nutrients (e.g., lactate, ketone bodies, fatty acids, etc.) to reutilize their own constituents for energy balance and to “feed” cancer cells (the so-called “reverse Warburg effect”) [227]. CAFs have the ability to regulate the metabolic remodeling of cancer cells in multiple ways, and to promote tumor growth through self-metabolic reprogramming [228]. A recent study by Gong et al. showed that CRC-associated fibroblasts underwent lipidomic reprogramming and increased CRC cell migration by these lipid metabolites’ cross-talk [232]. Autophagy was also shown to be attenuated in fibroblasts co-cultured with p53-deficient HCT116*^sh p53^ cells* by *ATG2B*-inhibiting exosomes. Exosomes obtained from HT29 cells with *TP53* mutations showed a similar effect. In doing so, the inhibitory effect of exosomal miR-4534, derived from CRC cells, was demonstrated, which also inhibited *ATG2B* in CAFs. Thus, the loss of *TP53* function may be responsible for impaired autophagy in CRC-associated CAFs [233].

Studies using the co-culture of human colorectal fibroblasts (CCD-18-co) and CRC cells (LoVo, SW480 and SW620) mimicking TME showed that fibroblasts positively influence CRC cell metabolism through the autophagy and oxidative stress pathway. An increase in the expression of autophagy-related proteins (e.g., LC3, BNIP3 and p62) was also demonstrated in CCD-18-co-cultured tumor cells. These fibroblasts could also accumulate vesicular structures, which were identified as autophagosomes. Furthermore, it has been shown that oxidative stress induced by CRC cells can stimulate autophagy in CAFs, which then leads to intensive glycolysis and attenuation of aerobic oxidation [234].

A recent interesting study has shown that colon carcinogenesis involves the proliferation of intestinal pericryptal leptin receptor (+) cells, which generate CAFs with a melanoma cell adhesion molecule (MCAM) (+) phenotype. They shape the immune microenvironment that promotes tumor growth. They interact with IL-1R to enhance NF-κB/IL-34/CCL8 signaling, which promotes macrophage chemotaxis [235].

IGF-1 was also confirmed to be an autocrine and paracrine inducer of CRC-associated fibroblast activation, which resulted in activation of IGF-1R/IR signaling. It was shown that IGF-1 secreted by irradiated CAFs’ binding to IGF-1R on CRC cells activated the AKT/mTOR pathway, causing glucose uptake and lactate release increasing solute carrier family 7 membrane 11 (SLC7A11) expression and promoting glutamine uptake by cancer cells [236].

CAFs are also a major source of IGF-2. This peptide was specifically induced in the CRC tumor stroma, and, through IGF-1R/IR signaling and paracrine and autocrine effects, activated pro-survival AKT signaling, enhanced tumor invasiveness and proliferative capacity and was responsible for increased local tumor regrowth after resection of the primary tumor. IGF-2 expression was shown to correlate with increased recurrence rates and poor survival in patients with CRC [237].

In summary, as in other TME cells, oxidative stress, hypoxia and other pro-autophagic factors induce autophagy in CAFs (Figure 3). Data on the direct involvement of the IGF system in the autophagy of these cells are lacking. The role of IGF-1 signaling is mainly explained by the mechanisms of increased proliferation, migration and cell survival via irradiated CAFs, which activate IGF-1R/AKT/mTOR signaling in cancer cells.

## 7. Autophagy and IGF System: Implications for Therapy in CRC

Autophagy is often compared to a double-edged sword in CRC therapy, as it can both lead to the death of cancer cells and promote their survival [27,32]. Regulatory mechanisms and inducing factors of autophagy, including small-molecule compounds (e.g., miRNAs, natural products, chemicals and radiation) in CRC cells are already presented in other papers [25,238,239]. The molecular mechanisms of CRC cells’ resistance to anti-EGFR mAbs [8] and to 5FU [240] are also described in detail, underlying the dysregulation of autophagy. It was shown that 5FU-resistant HCT-116 cells were able to overcome S-phase arrest and avoid apoptosis and activate autophagy [240]. Mechanisms of chemo-resistance to paclitaxel involved autophagy in CSCs, activated by the Cdx1/Bcl-2/LC3 pathway [208]. In planning therapeutic interventions, autophagy-specific proteins are also being considered. As an attractive target for the discovery of autophagy inhibitors, ATG4B has been identified as an important regulator of autophagy. A product named S130 could effectively inhibit the growth of xenografted HCT116 cells in vivo due to a defect in autophagy and increased stress sensitivity [241].

Potential CRC treatment strategies using knowledge of autophagy and IGF-1R signaling interactions are increasingly emerging [26,27,191,202,242,243,244]. In an in vitro model (HCT116 cells), it has been demonstrated that the use of branched chain AA (BCAA) supplementation significantly reduced insulin-initiated CRC cell proliferation. BCAA supplementation caused a marked reduction in the expression of activated IGF-1R and significantly increased autophagy (increased expression of LC3-II and BECN1) [242].

Natural plant agents have also been shown to be inducers of autophagy and apoptosis [243,244,245,246]. This group of autophagy inducers in vitro (HCT116 and HT29 cells) and in vivo includes Salvianolic Acid B (Sal B), an active compound extracted from the Chinese herb *Salvia miltiorrhiza*. As a single agent, it exerts its anti-cancer activity through suppression of the AKT/mTOR pathway. Transfection with the AKT plasmid resulted in AKT overexpression or pretreatment with insulin-reduced Sal B-induced autophagy and cell death [243]. Another agent is salidroside, a phenylpropanoid glycoside extracted from *Rhodiola rosea*. It exerted potent anti-proliferative properties in human CRC cells via a mechanism of PI3K/AKT/mTOR signaling inhibition [244]. On the other hand, chaetocochin J, an agent from the epipolythiodioxopiperazine alkaloids group, also exhibited anti-proliferative and pro-apoptotic properties. It enhanced autophagic flow in CRC cells via AMPK (activation) and PI3K/AKT/mTOR signaling (inhibition) [246]. Similar in its mechanism of action on autophagy is licoricidin (LCD) [245]. The anti-diabetic drug metformin (1,1-dimethylbiguanide hydrochloride) has also been shown to reduce CRC cell proliferation and migration by inducing cell cycle arrest in the G0/G1 phase. This was accompanied by a sharp decrease in c-Myc expression and downregulation of IGF-1R. The mechanisms of action involved the activation of AMPK and increased production of ROS, which inhibited the mTOR pathway [247].

Autophagy is also induced by the aforementioned anti-EGFR mAbs (Cetuximab) by inhibiting the class I PI3K/AKT/mTOR signaling, but not the MAPK/ERK pathway. Cetuximab leads to the promotion of association between beclin-1 and 3-MA (hVps34), which was inhibited by the overexpression of Bcl-2. Autophagy would protect cancer cells from the pro-apoptotic effects of this agent [248]. Similarly, justicidin has an autophagy-inducing effect in HT29 cells, acting by converting the autophagy marker LC3-I to LC3-II. Involved in this process are class III PI3K and Atg5 [249]. It has been shown that aspirin can also promote CRC cell autophagy through inhibition of the PI3K/AKT/Raptor pathway, and this involves the gene-encoding catalytic subunit p110α (PIK3CA)-mutated CRC cells [250].

A recent study was concerned with the possibility of inducing autophagy in CRC by a novel oncolytic agent, namely recombinant Chinese measles virus vaccine strain Hu191 (rMV-Hu191), which also regulated PI3K/AKT signaling [251]. In HCT116 cells and in a mouse model, the induction of autophagy was demonstrated after treatment with W922, and this was also in the mechanism of inhibition of the PI3K/AKT/mTOR pathway [252].

In contrast, Sipos et al. demonstrated different effects of IGF-1R and TLR9 signaling on CRC cell proliferation (HT-29 cells). This study showed that the concomitant application of tumor-derived self DNA with IGF-1R inhibitors exhibited an anti-proliferative effect, which was counteracted by the simultaneous inhibition of TLR9 signaling. The different effects of IGF-1R (picropodophyllin), TLR9 (ODN2088) and autophagy (chloroquine) inhibitors (per se or in combination) on HT29 cells suggested that the IGF-1R-related or non-IGF-1R-related autophagy process, however, also had “Janus-faced” characteristics in terms of its effect on CRC cell proliferation. The autophagy process induced by different combinations of self-DNA and inhibitors did not prevent HT-29 cell death but yielded the survival of some CD133+ HT-29 cells with stem-like cell features and can be the cause of CRC recurrence [27].

The authors proposed a model for bidirectional modulation of the autophagy pathway dependent on IGF-1R signaling. Targeting IGF-1R through suppression of the canonical PI3K/AKT/mTORC1 pathway stimulated the autophagy process. IGF-1R depletion inhibited mTORC2 which resulted in decreased PKCα/β activity, and consequently negatively affected autophagosome precursor formation [26,92].

Among the best-known inhibitors of autophagy in CRC cells are the following antimalarial drugs, chloroquine (CQ) and hydroxychloroquine (HCQ) (lysosome inhibitors), which reduce autophagy in the late stage and increase the susceptibility of cells to chemotherapy [253]. A group of these drugs passively diffuse into lysosomes, increasing their pH, disrupting their function and preventing the completion of autophagy [254]. Blocking autophagy with CQ has been shown to enhance the ability of CRC cells to induce the maturation of DCs, which can stimulate a T cell response, and this can result in an anti-tumor response [255]. Other inhibitors of autophagy in CRC are 3-MA [191,256] and a mold metabolite, wortmannin [191,254]. Both these factors are pan-PI3K inhibitors and are part of the so-called early-stage inhibitors of the autophagy machinery [254]. A recent study has shown that the inhibition of autophagy by 3-MA increased Twist1 expression in CRC cells and promoted the migration and invasion of cancer cells through the EMT process, thus having an adverse effect on CRCs [256]. IGF-1 alone was also shown to inhibit autophagy in HCT-8 cells resistant to 5-FU. After IGF-1 treatment, the number of autophagic bodies decreased. IGF-1 activated AKT/mTOR signaling, which inhibited autophagy. After the inhibition of autophagy, the drug-resistant cells became susceptible to apoptosis induced by 5-FU [202]. In addition, there are a number of synthetic PI3K pathway inhibitors and drugs that are undergoing clinical trials in various stages of the CRC [254].

Among the plant-derived autophagy inhibitors that have shown anti-cancer effects is a bicyclic hexapeptide glucoside isolated from *Rubia yunnanensis* called RA-XII. RA-XII treatment inhibited autophagosome formation. It was indicated that RA-XII suppressed autophagy by regulating several signaling pathways, including mTOR and NF-κB signaling. It was further shown that RA-XII could increase the sensitivity of CRC cells to bortezomib [257].

According to a recent review, more important than understanding the inducers and inhibitors of autophagy in cancer therapy is the identification of transcriptional regulators of this process, the evaluation of transcription factors as regulators of autophagy in response to oncogenic stress and the therapeutic implications of these data [258]. Some findings provide a rationale for chemotherapy targeting the PI3K/mTOR signaling pathway, providing a potential therapeutic strategy to enhance the efficacy of a dual PI3K/mTOR inhibitor in combination with an autophagy inhibitor for the treatment of CRC [259].

Currently, there are approximately 30 clinical trials actively investigating the activity of autophagy modulators in enhancing the efficacy of cytotoxic chemotherapy. A list of these clinical trials describing the results of CQ and HCQ in various cancers, but not involving CRC, is presented in another paper [260].

A summary of autophagy-modulating agents based on IGF-1R signaling regulation as potential chemotherapeutic agents in CRC discussed in this article is presented in Table 3.

## 8. Concluding Remarks and Future Directions

The presence of autophagy/mitophagy in most normal mature intestinal epithelial cells has been confirmed, where it has mainly protective functions against ROS-induced DNA and protein damage, and regulates proliferation, metabolism, differentiation, secretion of mucins and anti-microbial peptides (anti-tumoral autophagy role). Abnormal autophagy in intestinal epithelial cells leads to dysbiosis, a decline in local immunity and a decrease in cell secretory function. As a result, they may also lead to increased colonic SC proliferation or activation of pro-cancerogenic signaling pathways (Table 1). In general, therefore, autophagy prevents various diseases, including colonic disorders. Defects in autophagy are found in patients with MetS, IBD and CRC. Abnormalities in the IGF-1/IGF-1R signaling pathway in colonic epithelial cells lead to the dysregulation of intestinal homeostasis, which can also result in the development of precancerous lesions and CRC (Table 2). The main intracellular integrating center for autophagy-related signals is mTORC1. Under physiological conditions (sufficient presence of nutrients and growth factors, including insulin, IGFs, etc.), mTORC1 promotes cell growth and metabolic activity while suppressing the ULK1 complex and autophagy. Under starvation or cellular stress, multiple signaling pathways inactivate mTORC1 activity. This both inhibits cell growth to reduce energy demand and induces autophagy to allow survival. Energy depletion in cells activates AMPK and inhibits mTORC1, thereby promoting autophagy.

In CRC, the effects of the complex interactions of autophagy and IGF-1/IGF-1R signaling are expressed more clearly at the stage of tumor progression, in neoplastically transformed cells. Under conditions of hypoxia, ER stress, nutrient deprivation and oxidative stress, AMPK activation occurs, which induces autophagy in CRC cells. Autophagy is then a survival mechanism for metabolically altered neoplastic cells. In addition, autophagy is a mechanism that facilitates tumor development and progression (pro-tumoral autophagy role). In CRC neoplastic cells, IGF-1R signaling modulates the autophagy pathway bidirectionally. The inhibition of the canonical PI3K/AKT/mTORC1 pathway by IGF-1R results in the induction of autophagy. The opposite effect (inhibition of autophagy) is possible with IGF-1R depletion, which inhibits mTORC2, sequentially PKCα/β, thus negatively affecting autophagosome precursor formation. IGF-2 LOI by promoting autophagy promotes the pluripotency of CSCs.

The process of autophagy in CRC-associated adipocytes (CAAc) involves cell differentiation and is also under the control of insulin/IGF-1/PI3K/AKT/mTOR signaling. Cellular metabolic perturbations, oxidative stress, hypoxia, etc., also contribute to the increase in autophagy in cancer-associated fibroblasts (CAFs). In addition to CAFs and CAAs, more research is needed to assess the role of autophagy in tumor-associated immune cells (TIICs) and endothelial cells (ECs) during the different stages of colitis-associated CRC, taking into account obesity and insulin resistance. Understanding the role of IGF-1R signaling in these processes may also contribute to the development of new strategies for combination therapy in diabetes, colitis and colitis-associated CRC.

Given the dual role of autophagy in different phases of cancer, it is not surprising that both activators and inhibitors are possible anti-cancer agents. In addition, in CRC therapy, both the activation and inhibition of autophagy may improve treatment. Therapies based on IGF-1R inhibition or PI3K/mTOR inhibition in combination with autophagy-disrupting agents are suggested to block autophagy and thereby reduce neoplastic cell proliferation. By inhibiting IGF-1R signaling, the autophagy process is also inhibited, but precautions must be taken due to the variety of biological effects mediated by this signaling pathway [92].

The challenge of the future is to learn more about the role of autophagy in regulating the immune response in CRC involving TIICs to determine whether it is better to use agents that inhibit or stimulate autophagy. Further research is required to determine the therapeutic benefits of combination therapy, consisting of inhibitors of different components of IGF-1R signaling, autophagy and/or TLR9 signaling [27]. For a full evaluation of the role of ATG genes in CRC via the IGF system and as potential therapeutic targets in this cancer, it would also be important to continue research into the role of epigenetic factors that regulate the expression of these genes, and factors such as non-coding RNAs (e.g., miRNAs and lncRNAs) [261,262]. Continued research into the precise interactions of IGF-1 signaling and autophagy is important, mainly with respect to new anti-inflammatory and anti-tumor therapies, especially for patients with MetS [26,92]. Developing strategies to prevent obesity (and indirectly CRC) by regulating insulin production and activating insulin/IGF signaling, and also by understanding the mechanisms of autophagy, is one of the most important challenges today.

## Figures and Tables

**Figure 1 ijms-24-03665-f001:**
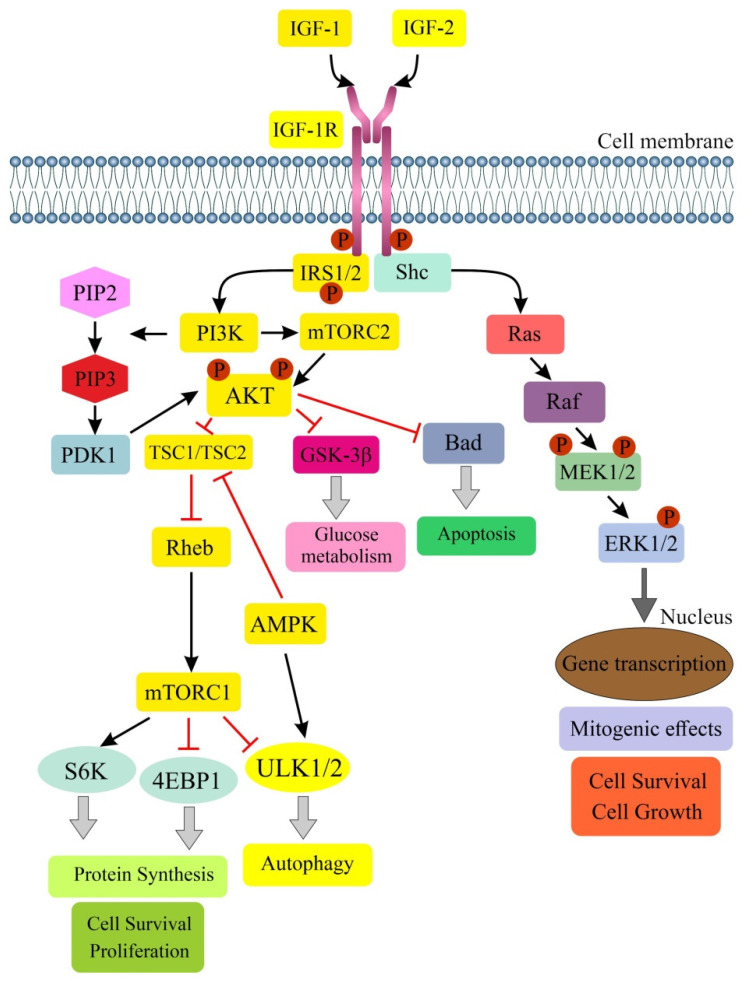
Schematic illustration of IGF/IGF-1R signaling pathway and its main downstream effectors. Stimulation of serine/threonine kinase AKT (named also PKB) activates mTORC1, leading to increase in protein synthesis. The Ras-Raf-ERK1/2 (MAPK) signaling is also activated to enhance mitogenic effects, cell survival and cell growth. IGF-1 binds mainly to IGF-1R, a typical tyrosine kinase membrane receptor and exerts growth-promoting, anti-apoptotic effects. Initial phosphorylation targets for IGF-1R include IRS 1/2, and among the downstream signaling effectors are PI3K, AKT, mTOR, S6K and Ras-ERK 1/2 (MAPK) pathway components. PI3K kinase, which has two subunits, p85 and p110, in turn phosphorylates a lipid protein, PIP2 to PIP3. PIP3 signals’ proteins such as PDK1 activate AKT by acting on its serine and threonine residues. Activated AKT inhibits GSK-3β and TSC1/TSC2 via phosphorylation. Inactive TSC1/TSC2 is unable to bind Rheb, which subsequently enables the activation of mTORC1 at the surface of lysosome, initiating its effect on S6K, 4EBP1 and ULK1/2, which are responsible for protein synthesis, cell survival, proliferation, delayed apoptosis and inhibition of autophagy. A potential stimulator of autophagy is AMPK signaling. The yellow boxes show the key components of IGF/IGF-1R and AMPK signaling in the regulation of autophagy, the details of which are discussed in the text. Legend: →: lead to; Ʇ: inhibits; AKT (PKB): serine/threonine kinase (protein kinase B); AMPK-5′ adenosine-monophosphate-activated protein kinase; Bad: Bcl-2 associated agonist of cell death; 4EBP1: eukaryotic translation initiation factor-binding protein 4E; GSK-3β: glycogen synthase kinase-3β; IGF: insulin-like growth factor; IGF-1R: IGF receptor type 1; IRS1/2: insulin-receptor substrate 1/2; MAPK: mitogen-activated protein kinase; mTORC1/2: mammalian target of rapamycin complex 1/2; PDK1-3: phosphoinositide-dependent protein kinase 1; PI3K: phosphatidylinositol-3-kinase; PIP2: phosphatidylinositol-4, 5-biphosphate; PIP3: phosphatidylinositol-3,4,5-triphosphate; Rheb: Ras homolog enriched in brain; shc: adaptor protein; S6K: S6 kinase; TSC1/2: tuberous sclerosis complex 1/2; ULK1/2: Unc-51-like autophagy activating kinase 1/2.

**Figure 2 ijms-24-03665-f002:**
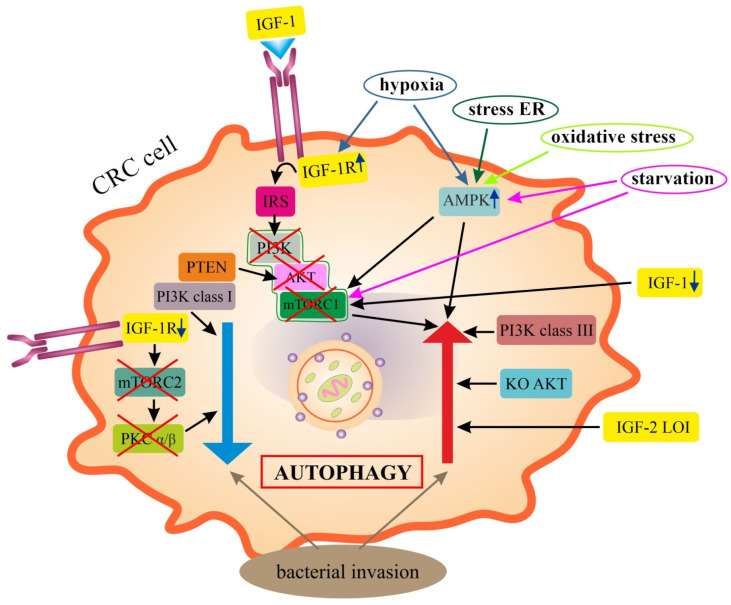
Schematic summary of the findings on the role of IGF system in the regulation of autophagy in colorectal cancer (CRC) cells. Neoplastic cells show both induction of autophagy (↑) and inhibition of autophagy (↓). Autophagy is promoted by IGF-1R, whose overexpression under hypoxia inhibits PI3K/AKT/mTORC1 signaling. Inhibition of mTORC1 also occurs under the impact of a decrease in IGF-1 and starvation. Autophagy is directly enhanced by class III PI3K, knockout of AKT and IGF-2 LOI. Inhibition of autophagy in CRC cells is mediated by IGF-1R depletion, through inhibition of mTORC2 and PKC α/β, as well as class I PI3K. Bacterial invasions of the colonic mucosa can both stimulate and inhibit autophagy. Legend: ↑/↓: increase, stimulation/decrease, inhibition; →: lead to; AKT: serine/threonine kinase; AMPK: 5′ adenosine-monophosphate-activated protein kinase; CRC: colorectal cancer; IGF-1/2: insulin-like growth factor 1/2; IGF-1R: IGF receptor type 1; IGF-2 LOI-IGF-2 loss of imprinting; IRS: insulin-receptor substrate; ER: endoplasmic reticulum; KO AKT: knockout of AKT gene; mTORC1/2: mammalian target of rapamycin complex 1/2; PI3K class I or III: phosphatidylinositol-3-kinase class I or III; PKC α/β: protein kinase C alpha/beta; PTEN: phosphatase and tensin homolog deleted on chromosome ten.

**Figure 3 ijms-24-03665-f003:**
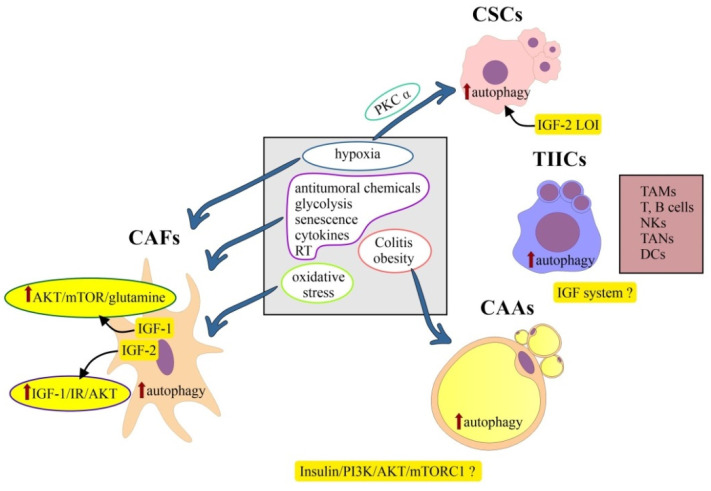
Simplified overview of the findings on the pro-autophagic factors (central box) and the potential role of insulin/IGF/PI3K/mTOR signaling (yellow boxes) in the regulation of autophagy in tumor microenvironment cells of colorectal cancer. Detailed mechanisms of regulation of autophagy involving the IGF pathway in TIICs and CAAs are the least understood. Role of IGF-1 in CAFs is mainly explained by the mechanisms of increased proliferation, migration and cell survival of cancer cells via irradiated CAFs, which activate AKT/mTOR/glutamine signaling in these cells. Legend: ↑: increase/stimulation; ?: unknown role; →: lead to; AKT: serine/threonine kinase; AMPK: 5′ adenosine-monophosphate-activated protein kinase; CAAs: cancer-associated adipocytes; CAFs: cancer-associated fibroblasts; CSCs: cancer stem cells; DCs: dendritic cells; IGF-1/2: insulin-like growth factor 1/2; IGF-2 LOI-IGF-2 loss of imprinting; IR: insulin receptor; NKs: natural killer cells; mTOR(C1): mammalian target of rapamycin (complex 1); PI3K: phosphatidylinositol-3-kinase; PKC α: protein kinase C alpha; RT: radiotherapy; TAMs: tumor-associated macrophages; TANs: tumor-associated neutrophils; TIICs: tumor-infiltrating immune cells.

**Table 1 ijms-24-03665-t001:** Cell-specific normal and impaired functions of autophagy in the colonic epithelial cells.

Cell Type	Autophagy	
	Normal Function	Effects of Functional Impairments
**Absorptive cell (colonocyte)**	cell differentiation cell survival ↑ cell metabolism ↔	dysbiosis ○ immune cell function ↓ ○ inflammation ↑ ○ autophagy Ʇ ATG genes/proteins ↓ ○ autophagy Ʇ ■ dysbiosis ■ cell permeability ↑ ■ cell proliferation ↑↑ ■ T cell infiltration ↑ autophagy ↑ ○ cell proliferation ↓
**Goblet cell**	mucin accumulation and secretion normal gut flora ↔ antimicrobial peptide production protection against colitis (*ATG7*) role in GAPs	autophagy ↓ ○ abnormal microflora ↑ ○ antimicrobial peptide ↓ ○ MUC-2 production ↓ ○ mucus ↓ ○ colitis sensitivity ↑
**Intestinal (colonic) stem cell**	cell maintenance controlling cell proliferation regulating cell differentiation regulating cell reprogramming generating PSCs	mitophagy ↑ (NOD2, ATG16L1) colitis ↑ (Slit2/Robo1 pathway) colonic SC proliferation ○ epithelial stress and cell hyperplasia Ʇ autophagy → EGFR/pMAPK/ERK ↑
**Paneth-like cell**	facilitating secretory function secretion of antibacterial products (e.g., lysozyme) preventing bacterial adhesion encoding niche factors supporting ISCs nutritional status detector	secretory autophagy ↑ ATG gene mutations (e.g., *ATG16L1*, *ATG5*, *ATG7*) ○ autophagy Ʇ ■ abnormal/reduced granules ↑ ■ abnormal exocytosis of granules ■ bacterial clearance ↓

Legend: ↑/↓: increase/decrease; stimulation/inhibition; ↔: maintenance; Ʇ: suppression; →: lead to; ATG: autophagy-related; EGFR: epidermal growth factor receptor; ERK: extracellular signal-regulated kinase; GAPs: goblet-cell-associated antigen passages; GLP-2: Glucagon-like peptide 2; IGF: insulin-like growth factor; (I)SCs: (intestinal) stem cells; (p)MAPK: (phosphorylated) mitogen-activated protein kinase; NOD2: the nucleotide-binding oligomerization domain-containing protein 2; PSCs: pluripotent stem cells.

**Table 2 ijms-24-03665-t002:** Cell-specific normal and impaired functions of IGF-1 signaling in the colonic epithelial cells.

Cell Type	Paracrine/Autocrine IGF-1 Signaling	
	Normal Function	Effects of Functional Impairments
**Absorptive cell (colonocyte)**	cell proliferation ↑ cell regeneration ↑ cell apoptosis ↓ protein catabolism and energy consumption ↓ expansion of ISCs and secretory cells mucosal barrier immunity ↔ intestinal immune homeostasis ↔	dysbiosis ↑ translocation of bacteria ↑ cell regeneration in colitis ↑ mucosal barrier restoration in colitis ↑ loss of intestinal homeostasis
**Goblet cell**	cell regeneration ↑ proliferation ↑	loss of intestinal homeostasis
**Intestinal (colonic) stem cell**	cell regeneration ↑ cell expansion SC stemness ○ SC renewal ○ pluripotency ○ differentiation	loss of intestinal homeostasis
**Paneth-like cell**	epithelial cell protection and repair ↑ mucosal inflammatory responses ↓ intestinal tropic effect of GLP-2 ↑	unknown

Legend: ↑/↓: increase/decrease; stimulation/inhibition; ↔: maintenance; GLP-2: glucagon-like peptide 2; IGF: insulin-like growth factor; (I)SCs: (intestinal) stem cells.

**Table 3 ijms-24-03665-t003:** List of chemotherapeutic strategies based on autophagic activity associated with IGF/IGF-1R signaling in colorectal cancer.

Autophagic Activity	Agents	Mechanism of Action	References
**Inducers**	Cetuximab	Dual inhibitor of EGFR and class I PI3K/AKTt/mTOR	[248]
	Adiponectin	Activator of AMPKα and PPARα and inhibitor of IGF-1/PI3K/AKT/mTOR	[200]
	JA	Increase expression of class III PI3K and Atg5	[249]
	SAL B	Inhibitor of AKT/mTOR	[243]
	Salidroside	Inhibitor of PI3K/AKT/mTOR	[244]
	LCD	Activator of AMPK and inhibitor of AKT/mTOR	[245]
	Metformin	Activator of AMPK and inhibitor of mTOR and IGF-1R	[247]
	Calycosin	Activator of SIRT1/AMPK and inhibitor of AKT/mTOR	[88]
	Aspirin	Inhibitor of PI3K/AKT/Raptor	[250]
	CJ	Activator of AMPK and inhibitor of PI3K/AKT/mTOR	[246]
	rMV-Hu191	Regulator of PI3K/AKT	[251]
	W922	Inhibitor of PI3K/AKT/mTOR	[252]
**Inhibitors**	3-MA	Inhibitor of class III PI3K	[191,256]
	Wortmannin	Inhibitor of PI3K	[191,254]
	IGF-1	Activator of AKT/mTOR in 5-FU-resistant cells	[202]
	RA-XII	Regulator of mTOR and NF-κB	[257]

Legend: AKT: serine/threonine kinase; AMPKα: 5′ adenosine-monophosphate-activated protein kinase α; Atg: autophagy-related genes/proteins; CJ: chaetocochin J; EGFR: epidermal growth factor receptor; 5-FU: 5-fluorouracil; JA: justicidin A; LCD: licoricidin; 3-MA: 3-methyladenine; mTOR: mammalian target of rapamycin; NF-κB: nuclear factor-kappa B; PI3K: phosphatidylinositol-3-kinase; PPARα: peroxisome proliferator-activated receptor α; Raptor: regulatory-associated protein of mTOR; rMV-Hu191: recombinant measles virus vaccine strain Hu191; Sal B: salvianolic acid B; SIRT1: sirtuin 1; RA-XII: bicyclic hexapeptidic glucoside from *Rubia yunnanensis.*

## Data Availability

Not applicable.

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
