# Peer review of "Autophagy and the Insulin-like Growth Factor (IGF) System in Colonic Cells: Implications for Colorectal Neoplasia"

_ijms, 2023, doi:10.3390/ijms24043665_

Round 1

Reviewer 1 Report

The author of the paper “Autophagy and the insulin-like growth factor (IGF) system in colonic cells: implications for colorectal neoplasia” have reviewed the role of IGF system in the molecular mechanisms of autophagy in the normal colon mucosa and in colorectal cancer. 

The author provides an extensive description of the state of the art and of current information about clinical implications. The manuscript overall is well organized and  detailed. 

Minor criticisms:

- I suggest adding a paragraph discussing how Insulin-like growth factor-binding proteins (IGFBPs) have also been implicated in colorectal cancer and autophagy mechanisms. 

- Moreover in the concluding paragraph, it would be appropriate to add a few comments on future prospects for this topic. 

Overall this review is of interest and well written. 

Author Response

Open Review

Dear Reviewer,

I wish to thank you very much for a favourable review, all critical remarks and time spent on reviewing the manuscript.

All changes (and all additions) in the text were marked in blue color.

Minor criticisms:

- I suggest adding a paragraph discussing how Insulin-like growth factor-binding proteins (IGFBPs) have also been implicated in colorectal cancer and autophagy mechanisms. 

I did not write this paragraph because there are too few data on the involvement of IGFBPs in autophagy in CRC. I have found only 3 papers (including one RETRACTED) on IGF2BP2 (attached). These findings are interesting and address the biological mechanisms of action of IGF2BP2 and selected roles of lncRNAs in CRC, also in association with autophagy, which I quoted in the last subsection without discussing.

  1. Wang Y, Lu JH, Wu QN, et al. LncRNA LINRIS stabilizes IGF2BP2 and promotes the aerobic glycolysis in colorectal cancer. Mol Cancer. 2019 Dec 2;18(1):174. doi: 10.1186/s12943-019-1105-0.
  2. Xia C, Li Q, Cheng X, et al. Insulin-like growth factor 2 mRNA-binding protein 2-stabilized long non-coding RNA Taurine up-regulated gene 1 (TUG1) promotes cisplatin-resistance of colorectal cancer via modulating autophagy. Bioengineered. 2022 Feb;13(2):2450-2469. doi: 10.1080/21655979.2021.2012918.
  3. retracted

- Moreover in the concluding paragraph, it would be appropriate to add a few comments on future prospects for this topic. 

 In accordance with the reviewer's recommendation, I have modified this text by adding a little more comments.

Overall this review is of interest and well written. 

Reviewer 2 Report

The author proposes in this review an enormous revision of literature concerning the role of autophagy and insulin-like growth factor (IGF) system in developing gut cancer.

The author has made a remarkable effort to choose the literature and to produce this revision, unfortunately some parts of the text are poorly organized and more than a description of the evidence seems a list of phrases.

I did a review by checking reference by reference and relative text up to page 6 of the 30 of text. Here in these I found about 15-20% of sentences copied in part or totally from the reference quotation or from someone nearby. This is very serious, so if the author does not want to be rejected the work will need a thorough revision of the whole text and a subsequent submission. (Note well, I stopped because it would have been a huge job without the software that control plagiarism. And then because I think it is incorrect on the part of the author submit a manuscript not highly accurate.)

I transformed the PDF into Word and added line numbers, so the author will find errors to correct with the line number.

Plagiarism highlighted:

Lanes 60-61, sentence of reference “3”: I found same words of this sentence (IGF-1 and IGF-1R are known to induce cell proliferation) to the abstract of reference 3 (Only 1% of the total serum IGF-1 is free and bioactive, and 80% of it binds to IGFBP-3. IGF-1 and its receptor IGF-1R are known to induce cell proliferation. Both IGF-1 and IGFBP-3 can favour angiogenesis by increasing the transcription of the VEGF gene.)

Lanes 61-63, sentence of references “4,5,6”: I found similar words of this sentence (In addition, the IGF system has been implicated in the development of resistance to both chemotherapeutic drugs and agents targeting the epidermal growth factor receptor (EGFR)) to the abstract of reference 4 (…the insulin receptor (IR), the IGF-1 receptor (IGF-1R), and IR substrate proteins 1 and 2 contribute to the transformation of normal colon epithelial cells. Moreover, the insulin/IGF system is also implicated in the development of resistance to both chemotherapeutic drugs and epidermal growth factor receptor targeted agents. The identification of hybrid receptors comprising both the IR and IGF-1R adds further complexity to this signaling network.).

Lanes 66-68, sentence of references “7,8,9”: I found similar words of this sentence (An important mechanism of mCRC resistance to anti-EGFR monoclonal antibody treatment (e.g., Cetuximab and Pani-tumumab) is dysregulation of cell autophagy) to the main text of reference 9 (Others

Autophagy and cancer stem cells (CSCs) also contribute to resistance to EGFR target therapy [18, 137]. Treatment with anti-EGFR agents results in dysregulation of autophagy [138]. Increased levels of autophagy-related proteins such as Beclin-1 and LC3 were observed in cetuximab-treated patients [139, 140].).

Lane 69-71, sentence of reference “10,11,12”: I found similar words of this sentence (Autophagy as a type II programmed cell death is a homeostatic, catabolic, multistep process of degradation of protein/DNA aggregates and cellular organelles that are engulfed by autophagosomes, digested in lysosomes and utilized to sustain cellular metabolism) found in the beginning of the abstract of reference 14 (I found similar sentence although this reference is forward) (Autophagy is a homeostatic, catabolic degradation process whereby cellular proteins and organelles are engulfed by autophagosomes, digested in lysosomes, and recycled to sustain cellular metabolism. Autophagy has dual roles in cancer, acting as both a tumor suppressor by preventing the accumulation of damaged proteins and organelles and as a mechanism of cell survival that can promote the growth….)

Lanes 72-73, sentence of reference “10, 13”: I found similar words of this sentence (Is a surveillance mechanism that protects cells from cancerous transformation also by reducing reactive oxygen species (ROS)) found in the text of reference 13 (Autophagy is a surveillance mechanism used by normal cells that protects them from the transformation to malignancy by removing damaged organelles and aggregated proteins and by reducing reactive oxygen species, mitochondrial abnormalities and DNA damage. However, autophagy…..)

Lanes 197-198, sentence of reference “65”: I found similar words of this sentence (IGF-1/IGF-1R signaling has also been shown to induce tumor-associated lymphangiogenesis and contribute to CRC lymphatic metastasis) found in the abstract conclusion of reference 65 (CONCLUSION: IGF-1/IGF-1R signaling induces tumor-associated lymphangiogenesis and contributes to lymphatic metastasis of CRC.)

Lanes 198-199, sentence of reference “66”: I found similar words of this sentence (Higher IGF-1R expres- sion is also associated with poorer response of rectal cancers to radiotherapy) found in the title of reference “66” (Insulin-like growth factor receptor-1 overexpression is associated with poor response of rectal cancers to radiotherapy.).

Lanes 202-205, sentence of reference “67”: I found similar words of this sentence (In vitro, chemotherapy-resistant cell lines show significantly higher levels of IGF-1R expression in nuclear compartment, and the protein inhibitor of activated signal transducer and activator of transcription 3 (STAT3, PIAS3) contributes to IGF-1R nuclear sequestration) found in the abstract of reference “67” (In vitro, chemoresistant cell lines presented significantly higher levels of IGF-1R expression within the nuclear compartment, and PIAS3, a protein implicated also in the sumoylation process of intranuclear proteins, contributed to IGF-1R nuclear sequestration, highlighting the essential role of PIAS3 in this process.

Lanes 205-207, sentence of reference “68”: I found similar words of this sentence (In vitro studies indicate that overexpression and activation of IGF-1R can increase the degree of transformation and motility of CRC cells through activation of c-Src) found in the abstract of reference “68” (Based on the above observations, we conclude that an overexpressed and activated IGF1-R may increase the degree of transformation and motility of colon cancer cells by activating c-Src.)

Lanes 207-208, sentence of reference “69”: I found similar words of this sentence (and/or requires the involvement of c-Met and the uPA/uPAR system) found in the title of reference “69” (Insulinlike growth factor-I-mediated migration and invasion of human colon carcinoma cells requires activation of c-Met and urokinase plasminogen activator receptor.)

Lanes 208-210, sentence of reference “70”: I found similar words of this sentence (The use of anti-IGF-1R monoclonal antibodies (MAbs), alone and in combination with oxaliplatin, led to a significant increase in cell apoptosis and significant inhibition of tumor cell proliferation and angiogenesis) found in the abstract of reference “70” (Anti-IGF-IR MoAB, alone and in combination with oxaliplatin, led to a significant increase in tumor cell apoptosis, and a significant inhibition of tumor cell proliferation and angiogenesis. Conclusions: These findings suggest that IGF-IR is a potential target for therapy in patients with advanced CRC.)

Lanes 210-213, sentence of reference “71”: I found similar words of this sentence (Similarly, a recent multicenter study in IBD patients showed that long-term treatment (54 weeks) of adult CD patients with Adalimumab (mAbs anti-TNF-α) results in a reduction in serum IGF-1 levels without changes in IGF BP4) found in the abstract of the reference “71” (We demonstrate that long-term treatment (54 weeks) of adult CD patients with adalimumab (ADA) results in a decrease in serum IGF-1 without changes in serum IGF-1 binding protein (IGF1BP4). These results prompted us to conduct a preclinical study to test the efficiency of IGF-1 in the medication for experimental colitis.)

Lanes 265-268, sentence of the reference “80”: I found similar words of this sentence (In the progression of CRC involving the IGF system, IGF/STAT3/NANOG/Slug signaling functions are additionally crucial by affecting the epithelial-mesenchymal transition (EMT) and the properties of cancer stem cells (CSCs)) found in the abstract/main text of the reference “80” (our data define the crucial functions of IGF/STAT3/NANOG/Slug signaling axis in the progression of CRC by operating EMT and CSCs properties, which make them served as potential therapeutic targets for treatment of CRC. FOUND IN THE ABSTRACT Our data suggest that the IGF/STAT3/NANOG/Slug signaling axis is an important pathway involved in both promoting EMT and maintaining CSCs in CRC. FOUND IN THE MAIN TEXT.)

Lanes 268-269, sentence of the reference “81”: I found similar words of this sentence (In addition, IGF-1/IR signaling is thought to play a specific role in the development of CRC in patients with T2DM) found in the abstract of the reference “81” (Higher expression of IGF-1, IGF-1R and IR proteins in CRC was associated with diabetes, suggesting IGF-1/IR signaling may play a special part in development of CRC in patients with diabetes.)

Paragraph or part that seems list of sentences:

From lane 84 to lane 90 (before reference 21)

From lane 94 to lane 100 (before references 26,27)

From lane 106 to lane 116 (before reference 36)

From lane 121 to lane 137 (before reference 42)

From lane 316 to lane 320 (around reference 91)

From lane 327 to lane 339 (before reference 94)

From lane 389 to lane 396 (before reference 107)

From lane 796 to lane 807 (before references 175, 184)

Figures (I printed the two figures pages and I think that they need to be revised)

Figure 1

In general, the boxes are too small à please increase their area

In general, the arrows/lines are too small à please increase them

The phosphate “P” are too small, like the word of “cell membrane” and “nucleus” à please increase the font

Please remove the shade of the box color à insert full color

In particular when you write in black and use dark color, like dark green or dark blue, it is difficult to the final readers read the word à please modify with brighter color (Change box color: APOPTOSIS, MEK ½, S6K, 4EBP1, SHC, IGF-1R, IGF-2)

Remove the brackets from the word legend and insert before the first legend acronym: “LEGEND”

Use “Bio-Render” online program to modify.

Figure 2:

In general, this figure in better than figure 1, but you have to change some things:

Increase the area of the boxes within the cell cartoon

Please remove the shade of the box color à insert full color

Change color of dark box with brighter color

Increase the arrows size

Increase the font of “bacterial invasion”

Change the words: “hypoxia”, “stress ER”, “oxidative stress”, “starvation” in BOLD

Moreover, the color of above words’ arrows become confused with the color of the cell membrane, please change

Remove the brackets from the word legend and insert before the first legend acronym: “LEGEND”

Add one Figure 3 in page 23 with TME immune cell infiltration that consider the three types of autophagy.

It will help the readers to understand better all text about TME

Use “Bio-Render” online program to build the figure.

Tables:

Table 1:

I would the title of the table move under the table and before the legend

Please change the symbol * of bulleted lists with more visible symbol (for example full square    

or full circle)

Increase the size of the “arrows or other symbols”

I would change the cells type name in BOLD, or increase the font

Remove the brackets from the word legend and insert before the first legend acronym: “LEGEND”

Table 2:

I would the title of the table move under the table and before the legend

I would change the cells type name in BOLD, or increase the font

I would change the name of signaling in BOLD

Please change the symbol * of bulleted lists with more visible symbol (for example full square of full circle )

Increase the font of the “arrows or other symbols”

Remove the brackets from the word legend and insert before the first legend acronym: “LEGEND”

Table 3

I would the title of the table move under the table and before the legend

Remove the brackets from the word legend and insert before the first legend acronym: “LEGEND”

Increase the font of the BOLD words

Main text:

1. Please control all reference that you added in main text, there are many write like this: (reviewed in: [254]) à Please change and leave only the square brackets with the reference/s number.

2. Sentence to be change:

A. lanes 150-152, reference “48”: please change the sentence with a phrase with more details about the 11ATGs described in the reference

B. lanes 161-162, reference “51”: please change the sentence with this text: “To summarize the involvement of F. nucleatum in development of CRC, a detailed description is devoted to another review.”

C. lanes 163: please cancel “Another bacterium” with: “Among the bacterial species important in growth of CRC there are…”

D. lane 328: Insert “a bridge” sentence about SIRT6, before “In a model of human…”

E. lane 461, reference “127”: change “in the most” with “one of the most”

F. lane 535, reference “138”: remove the second “in”

G. lane 618, reference “116” above: remove the second “,” and remove the space before comma

3. References

A. I would move the reference 34 in brackets with reference 35

B. lane 147 need reference, please insert.

C. Insert reference in lane 254 after “(PTEN)”, the author should add the reference 26 of reference 75 of this revised review

D. Insert reference in lane 838, one in lane 840, one in lane 842

E. Add one more reference (or more) when the author writes: “Recent studies”, “Only a few papers”, “Other authors”, “There are controversies”, “subsequent studies”, “Only a few authors”, “some of the authors”, “Interesting studies”, the mean is that the author wrote in plural and she added only one reference. Please modify.

4. Font:

A. pages 16, from lane 742 to lane 750, modify the size font, seem smaller than before lanes.

B. pages 16, from lane 768 to lane 825 page 17 please reduce the size font, its higher that before and after text.

5. Number of Paragraphs

I read the Instruction for author but I didn’t see any advice concerning the number/s of paragraphs.

I would add the subparagraphs number (for example 5.4.1), I think it is more precise and easier for the readers.

I hope to have made significant improvements to the author review.

Have good work.

Author Response

Dear Reviewer,

I wish to thank you very much for the time spent on reviewing the manuscript, as well as the very thorough and honest review from which I learned a lot.

For this paper, I used mostly keywords such as AUTOPHAGY, INSULIN-LIKE GROWTH FACTOR SIGNALING, COLORECTAL CANCER (CRC), EPITHELIAL NORMAL and CANCER CELLS, and tried to search as many functional links as possible in this issue. It was also not my intention to present in detail issues that are already presented in other reviews, although I also cite many such works to condense issues already described.

All changes and/or additions that have been applied in the text are made and marked in red.

Plagiarism highlighted:

I have tried to cite content that is important for my review. This is very difficult, because in the literature of broad topics (CRC-autophagy-IGF axis), a lot of content, names, even definitions are often repeated and it is impossible to write some expressions differently. I am extremely sorry that the reviewer considered the retention of one sentence from the some abstracts of the papers [e.g. 3, 4] as plagiarism.

In addition, the exact citations of papers are used everywhere, which excludes plagiarism. Writing single sentences in inverted commas is rather not accepted in review papers. The main repeated words shown by the reviewer as plagiarized are the „key words” of this review (e.g., colon cancer cells, IGF-1/IGF-1R signaling, autophagy, CRC, rectal cancer, cell proliferation, chemotherapy, resistance, prognostic, transformation, motility, tumor-associated lymphangiogenesis), proper names (IGF, STAT3, NANOG, IGF/STAT3/NANOG/Slug, IGF-1R, CD), terms from the list of abbreviations of my work (c-Met and the uPA/uPAR system) or generally used sentences defining the results of scientific research (higher level of…, significant increase …in cell apoptosis, inhibiton of tumor cell proliferation, play a specific role, etc.), cited after other authors.

I have tried to correct the most underlined sentences, while preserving the essential content of the research, or the conclusions of other authors regarding the observed phenomena.

Lanes 60-61, sentence of reference “3”: I found same words of this sentence (IGF-1 and IGF-1R are known to induce cell proliferation) to the abstract of reference 3 (Only 1% of the total serum IGF-1 is free and bioactive, and 80% of it binds to IGFBP-3. IGF-1 and its receptor IGF-1R are known to induce cell proliferation. Both IGF-1 and IGFBP-3 can favour angiogenesis by increasing the transcription of the VEGF gene.)

In accordance with the reviewer's recommendation, the entire paragraph has been changed, retaining the relevant content from the works cited. The numbering of the citations was also changed and paper 3 was replaced with another paper more appropriate for this content of my review.

Lanes 61-63, sentence of references “4,5,6”: I found similar words of this sentence (In addition, the IGF system has been implicated in the development of resistance to both chemotherapeutic drugs and agents targeting the epidermal growth factor receptor (EGFR)) to the abstract of reference 4 (…the insulin receptor (IR), the IGF-1 receptor (IGF-1R), and IR substrate proteins 1 and 2 contribute to the transformation of normal colon epithelial cells. Moreover, the insulin/IGF system is also implicated in the development of resistance to both chemotherapeutic drugs and epidermal growth factor receptor targeted agents. The identification of hybrid receptors comprising both the IR and IGF-1R adds further complexity to this signaling network.).

In accordance with the reviewer's recommendation, the entire paragraph has been changed, retaining the relevant content from the works cited. The numbering of the citations was also changed and paper 3 was replaced with another paper more appropriate for this content of my review.

Lanes 66-68, sentence of references “7,8,9”: I found similar words of this sentence (An important mechanism of mCRC resistance to anti-EGFR monoclonal antibody treatment (e.g., Cetuximab and Pani-tumumab) is dysregulation of cell autophagy) to the main text of reference 9 (Others

Autophagy and cancer stem cells (CSCs) also contribute to resistance to EGFR target therapy [18, 137]. Treatment with anti-EGFR agents results in dysregulation of autophagy [138]. Increased levels of autophagy-related proteins such as Beclin-1 and LC3 were observed in cetuximab-treated patients [139, 140].).

There are papers that highlight the importance of autophagy as an important mechanism involved in resistance to EGFR-based agent therapy, and that is why it has been quoted here. By providing such information, it is impossible to completely change the words or their meaning.

In accordance with the reviewer's recommendation, the wording of the information quoted from the work has been changed, with the original meaning retained.

Lane 69-71, sentence of reference “10,11,12”: I found similar words of this sentence (Autophagy as a type II programmed cell death is a homeostatic, catabolic, multistep process of degradation of protein/DNA aggregates and cellular organelles that are engulfed by autophagosomes, digested in lysosomes and utilized to sustain cellular metabolism) found in the beginning of the abstract of reference 14 (I found similar sentence although this reference is forward) (Autophagy is a homeostatic, catabolic degradation process whereby cellular proteins and organelles are engulfed by autophagosomes, digested in lysosomes, and recycled to sustain cellular metabolism. Autophagy has dual roles in cancer, acting as both a tumor suppressor by preventing the accumulation of damaged proteins and organelles and as a mechanism of cell survival that can promote the growth….)

Lanes 72-73, sentence of reference “10, 13”: I found similar words of this sentence (Is a surveillance mechanism that protects cells from cancerous transformation also by reducing reactive oxygen species (ROS)) found in the text of reference 13 (Autophagy is a surveillance mechanism used by normal cells that protects them from the transformation to malignancy by removing damaged organelles and aggregated proteins and by reducing reactive oxygen species, mitochondrial abnormalities and DNA damage. However, autophagy…..)

Also in this section of the paper, the citation of definitions of autophagy is encyclopaedic information, repeated almost uniformly by many authors and necessarily appear similar. However, my citations are not identical as the reviewer suggests. In addition, exact citations of papers are used everywhere, which excludes plagiarism. Writing single sentences in inverted commas is also not accepted in review papers. However, the paragraph has been reworded as recommended.

Lanes 197-198, sentence of reference “65”: I found similar words of this sentence (IGF-1/IGF-1R signaling has also been shown to induce tumor-associated lymphangiogenesis and contribute to CRC lymphatic metastasis) found in the abstract conclusion of reference 65 (CONCLUSION: IGF-1/IGF-1R signaling induces tumor-associated lymphangiogenesis and contributes to lymphatic metastasis of CRC.)

In accordance with the reviewer's recommendation, the wording of the information quoted from the work has been changed, with the original meaning retained.

Lanes 198-199, sentence of reference “66”: I found similar words of this sentence (Higher IGF-1R expres- sion is also associated with poorer response of rectal cancers to radiotherapy) found in the title of reference “66” (Insulin-like growth factor receptor-1 overexpression is associated with poor response of rectal cancers to radiotherapy.).

In accordance with the reviewer's recommendation, the wording of the information quoted from the work has been changed, with the original meaning retained.

Lanes 202-205, sentence of reference “67”: I found similar words of this sentence (In vitro, chemotherapy-resistant cell lines show significantly higher levels of IGF-1R expression in nuclear compartment, and the protein inhibitor of activated signal transducer and activator of transcription 3 (STAT3, PIAS3) contributes to IGF-1R nuclear sequestration) found in the abstract of reference “67” (In vitro, chemoresistant cell lines presented significantly higher levels of IGF-1R expression within the nuclear compartment, and PIAS3, a protein implicated also in the sumoylation process of intranuclear proteins, contributed to IGF-1R nuclear sequestration, highlighting the essential role of PIAS3 in this process.

In accordance with the reviewer's recommendation, the wording of the information quoted from the work has been changed, with the original meaning retained.

Lanes 205-207, sentence of reference “68”: I found similar words of this sentence (In vitro studies indicate that overexpression and activation of IGF-1R can increase the degree of transformation and motility of CRC cells through activation of c-Src) found in the abstract of reference “68” (Based on the above observations, we conclude that an overexpressed and activated IGF1-R may increase the degree of transformation and motility of colon cancer cells by activating c-Src.)

In accordance with the reviewer's recommendation, the wording of the information quoted from the work has been changed, with the original meaning retained.

Lanes 207-208, sentence of reference “69”: I found similar words of this sentence (and/or requires the involvement of c-Met and the uPA/uPAR system) found in the title of reference “69” (Insulinlike growth factor-I-mediated migration and invasion of human colon carcinoma cells requires activation of c-Met and urokinase plasminogen activator receptor.)

In accordance with the reviewer's recommendation, the wording of the information quoted from the work has been changed, with the original meaning retained.

Lanes 208-210, sentence of reference “70”: I found similar words of this sentence (The use of anti-IGF-1R monoclonal antibodies (MAbs), alone and in combination with oxaliplatin, led to a significant increase in cell apoptosis and significant inhibition of tumor cell proliferation and angiogenesis) found in the abstract of reference “70” (Anti-IGF-IR MoAB, alone and in combination with oxaliplatin, led to a significant increase in tumor cell apoptosis, and a significant inhibition of tumor cell proliferation and angiogenesis. Conclusions: These findings suggest that IGF-IR is a potential target for therapy in patients with advanced CRC.)

In accordance with the reviewer's recommendation, the wording of the information quoted from the work has been changed, with the original meaning retained.

Lanes 210-213, sentence of reference “71”: I found similar words of this sentence (Similarly, a recent multicenter study in IBD patients showed that long-term treatment (54 weeks) of adult CD patients with Adalimumab (mAbs anti-TNF-α) results in a reduction in serum IGF-1 levels without changes in IGF BP4) found in the abstract of the reference “71” (We demonstrate that long-term treatment (54 weeks) of adult CD patients with adalimumab (ADA) results in a decrease in serum IGF-1 without changes in serum IGF-1 binding protein (IGF1BP4). These results prompted us to conduct a preclinical study to test the efficiency of IGF-1 in the medication for experimental colitis.)

In accordance with the reviewer's recommendation, the wording of the information quoted from the work has been changed, with the original meaning retained.

Lanes 265-268, sentence of the reference “80”: I found similar words of this sentence (In the progression of CRC involving the IGF system, IGF/STAT3/NANOG/Slug signaling functions are additionally crucial by affecting the epithelial-mesenchymal transition (EMT) and the properties of cancer stem cells (CSCs)) found in the abstract/main text of the reference “80” (our data define the crucial functions of IGF/STAT3/NANOG/Slug signaling axis in the progression of CRC by operating EMT and CSCs properties, which make them served as potential therapeutic targets for treatment of CRC. FOUND IN THE ABSTRACT Our data suggest that the IGF/STAT3/NANOG/Slug signaling axis is an important pathway involved in both promoting EMT and maintaining CSCs in CRC. FOUND IN THE MAIN TEXT.)

In accordance with the reviewer's recommendation, the wording of the information quoted from the work has been changed, with the original meaning retained.

Lanes 268-269, sentence of the reference “81”: I found similar words of this sentence (In addition, IGF-1/IR signaling is thought to play a specific role in the development of CRC in patients with T2DM) found in the abstract of the reference “81” (Higher expression of IGF-1, IGF-1R and IR proteins in CRC was associated with diabetes, suggesting IGF-1/IR signaling may play a special part in development of CRC in patients with diabetes.)

In accordance with the reviewer's recommendation, the wording of the information quoted from the work has been changed, with the original meaning retained.

Paragraph or part that seems list of sentences:

Unfortunately, the version sent for revisions after the reviews has no lines, nor could I rewrite it myself, hence this part of proofreading the work was very difficult. I apologise if I did not fully understand the intention of the reviewer's corrections.

From lane 84 to lane 90 (before reference 21)

In accordance with the reviewer's recommendation, the wording of the information quoted from the work has been changed, with the original meaning retained.

From lane 94 to lane 100 (before references 26,27)

The papers cited here [26,27] refer to one line on page 2, I cannot see lines from 94-100, and include 'my' one own sentence. The papers cited here are to make the reader aware that the role of IGF-1 axis in autophagy in CRC has been undertaken by few authors, hence the purpose of the current review.

As far as the reviewer was concerned, the most important content of papers 23, 24, 25 (before 26,27) what I meant here was to outline the subject matter of other review papers (as this is an introduction to the whole paper), concerning mechanisms related to autophagy in CRC that have already been presented. Longer descriptions of these papers, in my opinion, would be pointless, and to omit them would be at the loss of the reader wishing to expand their knowledge. 

From lane 106 to lane 116 (before reference 36)

In accordance with the reviewer's recommendation, I have modified this text by adding a little more details and modifying the cited literature. The abbreviation of this part of the whole subsection is due to the fact that the exact cellular mechanisms will be discussed later in the section on individual cells, and here it was only meant to signal the issues raised recently on autophagy in CRC from a clinical aspect.

From lane 121 to lane 137 (before reference 42)

In accordance with the reviewer's recommendation, I have modified this text by adding a little more details. This subsection is a snapshot of the link between genetic alterations in autophagy genes and CRC, as this is thoroughly presented in other reviews [32,38].

From lane 316 to lane 320 (around reference 91)

In accordance with the reviewer's recommendation, I have modified this text by adding a little more detail.

From lane 327 to lane 339 (before reference 94)

In accordance with the reviewer's recommendation, I have modified this text.

From lane 389 to lane 396 (before reference 107)

In accordance with the reviewer's recommendation, I have modified this text by adding a little more details and have moved one citation to the later part of the work [Antico S'2013, now is number 125 of references].

From lane 796 to lane 807 (before references 175, 184)

In accordance with the reviewer's recommendation, I have modified this text by adding a little more detail and modifying the cited literature. I have also renumbered items 175 to 176, as this was a mistake. It was about the works [176,184] in this paragraph of the work. I apologise for this mistake.

Figures (I printed the two figures pages and I think that they need to be revised)

Figure 1

In general, the boxes are too small à please increase their area

In general, the arrows/lines are too small à please increase them

The phosphate “P” are too small, like the word of “cell membrane” and “nucleus” à please increase the font

Please remove the shade of the box color à insert full color

In particular when you write in black and use dark color, like dark green or dark blue, it is difficult to the final readers read the word à please modify with brighter color (Change box color: APOPTOSIS, MEK ½, S6K, 4EBP1, SHC, IGF-1R, IGF-2)

Remove the brackets from the word legend and insert before the first legend acronym: “LEGEND”

Use “Bio-Render” online program to modify.

I have corrected the things highlighted by the reviewer, but the size of the figure, and as for the size of the whole image for printing, readability can be improved by using the print options: ,,orientation'', ,,paper size, ,,image size, ,,page margins'', ,,fit'' etc.

Figure 2:

In general, this figure in better than figure 1, but you have to change some things:

Increase the area of the boxes within the cell cartoon

Please remove the shade of the box color à insert full color

Change color of dark box with brighter color

Increase the arrows size

Increase the font of “bacterial invasion”

Change the words: “hypoxia”, “stress ER”, “oxidative stress”, “starvation” in BOLD

Moreover, the color of above words’ arrows become confused with the color of the cell membrane, please change

Remove the brackets from the word legend and insert before the first legend acronym: “LEGEND”

In accordance with the reviewer's recommendation, I have corrected also Figure 2. I have removed the recommended brackets under the Figure description, although brackets are also used in this journal.

Add one Figure 3 in page 23 with TME immune cell infiltration that consider the three types of autophagy.

It will help the readers to understand better all text about TME

Use “Bio-Render” online program to build the figure.

In accordance with the reviewer's recommendation, I have added Figure 3 with the involvement of TME cells (in addition to CRC cells – Fig. 2) in autophagy and the known or as yet unknown effects of the IGF system on this process in these cells.

Tables:

Table 1:

I would the title of the table move under the table and before the legend

Please change the symbol * of bulleted lists with more visible symbol (for example full square ▪    or full circle●)

Increase the size of the “arrows or other symbols”

I would change the cells type name in BOLD, or increase the font

Remove the brackets from the word legend and insert before the first legend acronym: “LEGEND”

I have not changed the captions above the Tables, as they always have the headings at the top (unlike the figure descriptions) and the legends at the bottom. I have corrected what I could regarding the reviewer's recommendations. The size and bolding of the font is ultimately decided by the publisher in the final printing of the paper.

Table 2:

I would the title of the table move under the table and before the legend

I have not changed the captions above the Tables, as they always have the headings at the top (unlike the figure descriptions) and the legends at the bottom. I have corrected what I could regarding the reviewer's recommendations.

I would change the cells type name in BOLD, or increase the font

I have changed it. However, the size and bolding of the font is ultimately decided by the publisher in the final printing of the work....

I would change the name of signaling in BOLD

Please change the symbol * of bulleted lists with more visible symbol (for example full square ▪ of full circle ●)

I have changed it.

Increase the font of the “arrows or other symbols”

Remove the brackets from the word legend and insert before the first legend acronym: “LEGEND”

 I have changed it.

Table 3

I would the title of the table move under the table and before the legend

Thank you for this comment, but I have not changed the captions above the Tables, as they always have the headings at the top (unlike the figure descriptions) and the legends at the bottom. I have corrected what I could regarding the reviewer's recommendations.

The size and bolding of the font is ultimately decided by the publisher in the final printing of the work....

Remove the brackets from the word legend and insert before the first legend acronym: “LEGEND”

I have changed it.

Increase the font of the BOLD words

Main text:

Please control all reference that you added in main text, there are many write like this: (reviewed in: [254]) à Please change and leave only the square brackets with the reference/s number.

Thank you for this comment, I have corrected everywhere. However, I would like to mention that this spelling is also used, especially in cases of detailed studies that cannot be summarized in few words. And it would also be a pity not to quote these review papers in general.

  1. Sentence to be change:
  2. lanes 150-152, reference “48”: please change the sentence with a phrase with more details about the 11ATGs described in the reference

In accordance with the reviewer's recommendation, the data for publication 48 were completed.

  1. lanes 161-162, reference “51”: please change the sentence with this text: “To summarize the involvement of F. nucleatum in development of CRC, a detailed description is devoted to another review.”

In accordance with the reviewer's recommendation, the data for publication 51 were completed.

  1. lanes 163: please cancel “Another bacterium” with: “Among the bacterial species important in growth of CRC there are…”

In accordance with the reviewer's recommendation, the data for publication 52 were modified.

  1. lane 328: Insert “a bridge” sentence about SIRT6, before “In a model of human…”

In accordance with the reviewer's recommendation, a binding sentence was added before citing work [93].

  1. lane 461, reference “127”: change “in the most” with “one of the most”

In accordance with the reviewer's recommendation, it has been corrected.

  1. lane 535, reference “138”: remove the second “in”

In accordance with the reviewer's recommendation, I removed the second “in”

  1. lane 618, reference “116” above: remove the second “,” and remove the space before comma

I did not find such an error above reference [116], nor elsewhere (I checked 23 places using this mark). Without having lines in the paper, it is sometimes difficult to find errors noted by the reviewer.

  1. References
  2. I would move the reference 34 in brackets with reference 35

Paper 34 is cited twice and is about colitis-associated cancer i.e. its citations are in the right place. But the truth is that it can also be cited together with [35] and so I did.

  1. lane 147 need reference, please insert.

Without having lines in the paper, it is sometimes difficult to find errors noted by the reviewer.

I hope you were referring to this citation. i.e. [48] (on page 4). This paper was cited after more information.

  1. Insert reference in lane 254 after “(PTEN)”, the author should add the reference 26 of reference 75 of this revised review

Of course, I have added.

  1. Insert reference in lane 838, one in lane 840, one in lane 842

Without having lines in the paper from the editor, it is was difficult to find these places. I am sorry, but I could not insert the citations, but the citation is always after the presentation of a brief description of the work in question, so it will definitely be on.

  1. Add one more reference (or more) when the author writes: “Recent studies”, “Only a few papers”, “Other authors”, “There are controversies”, “subsequent studies”, “Only a few authors”, “some of the authors”, “Interesting studies”, the mean is that the author wrote in plural and she added only one reference. Please modify.

Thank you for this comment, I have corrected it.

  1. Font:
  2. pages 16, from lane 742 to lane 750, modify the size font, seem smaller than before lanes.

It is not my fault, my original work has no such changes in font sizes. I'm sorry, but it's a matter of uploading publications online through various systems and saving in other programs. Of course the publisher will correct everything during preparing the final version.

  1. pages 16, from lane 768 to lane 825 page 17 please reduce the size font, its higher that before and after text.

Similarly, it is not my fault, my original work has no such changes in font sizes. I'm sorry, but it's a matter of uploading publications online through various systems and saving in other programs. Of course the publisher will correct everything during preparing the final version.

  1. Number of Paragraphs

I read the Instruction for author but I didn’t see any advice concerning the number/s of paragraphs.

I would add the subparagraphs number (for example 5.4.1), I think it is more precise and easier for the readers.

Thank you for this comment, I have corrected it.

I hope to have made significant improvements to the author review.

Have good work.

It was hard at the appointed time, but thank you for all your comments. I apologise for the effort put into sentences that are not always clear, or the overly abbreviated treatment of works (papers), especially other opinion pieces or reviews.

Round 2

Reviewer 2 Report

The author proposes in this review an enormous revision of literature concerning the role of autophagy and insulin-like growth factor (IGF) system in developing gut cancer.

The author has made a remarkable effort to increase the level of this review following my advice.

I accept this manuscript, with minor revision.

There some grammar errors to be solve.

I'm asking to the journal to control if the sentences are original with purchase software.

Correct:
Lane 105: Authophagy in Autophagy

Lane 149: Macrophagy à do you mean Macroautophagy?

Lane 245: MAbs in mAbs

Lane 246: oxiplatin in oxaliplatin

Lane 296: protein kinase C (PKC) and serum and à protein kinase C (PKC), serum and

Lane 389: proces à process

Table 2: mucosal barier restoration in colitis à mucosal barrier restoration in colitis

Lane 725: remove the double “the”

Lane 741: the stady state à the steady state

Lane 832: remove space after strains

Lane 833: add space after TLR

Lane 836: neighbouring à neighboring

Lane 917: intergration à integration

Lane 1083: (as a AKT..) à (as an AKT..)

Lane 1239: immunsuppresion à immunosuppression

Lane 1341=1376: MoAbs à mABs

Table 3: Authophagic activity à Autophagic activity

Acronym: BNIP3L Bcl-2/adenowirus à adenovirus

Author Response

Dear Reviewer,

I wish to thank you very much for a favourable review, time spent on reviewing the manuscript, and demonstrating additional errors. These have been corrected in the current version of the paper. Thank you very much again for such an insightful review. All changes in the current text were marked in blue color.
